# Structural-disorder-driven critical quantum fluctuation and localization in two-dimensional semiconductors

Bong Gyu Shin[1,2,3], Ji-Hoon Park [4], Jz-Yuan Juo[1], Jing Kong [4] &
Soon Jung Jung [1]✉

Quantum fluctuations of wavefunctions in disorder-driven quantum phase transitions (QPT) exhibit criticality, as evidenced by their multifractality and power law behavior. However, understanding the metal-insulator transition (MIT) as a continuous QPT in a disordered system has been challenging due to fundamental issues such as the lack of an apparent order parameter and its dynamical nature. Here, we elucidate the universal mechanism underlying the structural-disorder-driven MIT in 2D semiconductors through auto-correlation and multifractality of quantum fluctuations. The structural disorder causes curvature-induced band gap fluctuations, leading to charge localization and formation of band tails near band edges. As doping level increases, the localization-delocalization transition occurs when states above a critical energy become uniform due to unusual band bending by localized charge. Furthermore, curvature induces local variations in spin-orbit interactions, resulting in non-uniform ferromagnetic domains. Our findings demonstrate that the structural disorder in 2D materials is essential to understanding the intricate phenomena associated with localization-delocalization transition, charge percolation, and spin glass with both topological and magnetic disorders.

Localization of charge and spin in a disordered semiconductor has been intensively investigated, since it profoundly affects charge or spin diffusion, band tails, quantum Hall transition, metal–insulator transition (MIT), and many other topics[1-21]. In particular, MIT is one of the unconventional quantum phase transitions, which has not been fully understood yet due to fundamental difficulties such as the lack of an obvious order parameter and complexity related to the coexistence of metallic and insulating states in an intermediate regime of the MIT[5,6,8-10]. Following the pioneering works of the "Anderson transition", the localization–delocalization transition (or MIT) in disordered systems has emerged as a continuous quantum phase transition accompanied by global symmetry and dimensionality[1,5,6,16]. Since the discovery of MIT in the two-dimensional (2D) electron gas in semiconductor devices, the scaling theory and renormalization group approaches based on the non-linear sigma model have been established to explain MIT in disordered 2D systems with interacting pictures[1-5,7,8,12-14,17,22-24]. This is in contrast to the prior prediction of the scaling theory, which anticipates that all states of 2D non-interacting systems are weakly localized and no true metallic states at 0 K exist[17]. To understand the MIT of disordered interacting electrons, an alternative approach based on the dynamical mean-field theory was constructed with local order parameters with the typical value of the local density of states (LDOS)[6,25,26]. A peculiar feature of disorder-driven MIT is the existence of quantum fluctuations of wavefunctions exhibiting

[1]Max Planck Institute for Solid State Research, Heisenbergstrasse 1, 70569 Stuttgart, Germany. [2]SKKU Advanced Institute of Nanotechnology (SAINT), Sungkyunkwan University (SKKU), Suwon 16419, Republic of Korea. [3]Department of Nano Science and Technology, Sungkyunkwan University (SKKU), Suwon 16419, Republic of Korea. [4]Department of Electrical Engineering and Computer Science, Massachusetts Institute of Technology, Cambridge, MA 02139, USA. ✉e-mail: s.jung@fkf.mpg.de

multifractal behavior at criticality, which is related to the scaling behavior of moments of LDOS with a system size[1-3]. Both approaches independently indicate that the LDOS is key to a complete understanding of MIT.

MIT in 2D materials is of particular interest due to both their versatility and the underlying exotic physics. Experimental observation of MIT in semiconducting TMdCs was reported via doping control in a field effect transistor device structure[11,15], in which insulating and metallic states were divided by the critical doping charge density. Confusingly, the MIT was not observed when the $MoS_2$ was encapsulated by hexagonal BN, contacted with aluminum electrodes, or investigated as an exfoliated sample, which implies a significant role played by extrinsic factors[27-29]. Furthermore, peculiar transport results in TMdCs were observed such as dominant charge trap and band tails near the band edge, which cannot be explained by in-gap states of defects[27,30,31]. However, the cause of the MIT and the localized charge trap in TMdCs was not explained fully. To explain these phenomena, and to elucidate their underlying physics, investigation at the microscopic scale is imperative. Indeed, direct observation of wavefunctions at the atomic scale can provide insight into localization and multifractality that is expected near or at the critical point of MIT as localization length diverges[1-3,32].

Here, we provide a universal mechanism of structural-disorder-driven MIT in 2D semiconductors based on gate-tunable scanning tunneling microscopy (STM) and spectroscopy (STS) results. In the randomly corrugated $MoS_2$ on $SiO_2$, the localization of the doping charge was confirmed by a spatial flattening of band edge and a modulation of the local tunneling barrier height (TBH). The direct evidence of the localization–delocalization transition in the structurally disordered monolayer was provided by the autocorrelation (localization length), distribution, and multifractality of wavefunctions near or at the critical point, showing the power law, the change from log-normal (insulating) to normal (metallic) distributions, and the singularity spectra towards the metallic limit of normalized STS mapping results, respectively[1-4,33-35]. In addition, STS and theoretical calculation results confirm that the structural disorder in TMdCs is the origin of the band tail. Moreover, the density functional theory (DFT) calculations confirmed that the bending strain induces localized magnetic moments correlated to doping charge localization. Surprisingly, noncollinear DFT calculations show non-uniform ferromagnetic domains in the structural disorder, known as spin glass. The values align well, both qualitatively and quantitatively, with previously reported experimental magnetization values[36].

The observed curvature effects of the structural disorder can explain many phenomena reported previously, such as formation of MIT, band tails, charge trap states near band edges, temperature-dependent percolation behaviors, negative magnetoresistance, and intrinsic magnetization[15,27,30,31,36-39]. Furthermore, the observed localization of charge and magnetic moment is not limited to $MoS_2$, and has been confirmed in other structurally deformed TMdCs.

## Results and discussion
### Structural-disorder-driven charge localization
Figure 1a–c shows a representative STM image of the randomly deformed monolayer $MoS_2$ (Fig. 1a) and the corresponding local curvature (Fig. 1b) and band gap (Fig. 1c). The structurally disordered monolayer $MoS_2$ conforms to the surface roughness of the $SiO_2$ substrate (Fig. 1a and Supplementary Fig. 1) and exhibits large curvature or bending strain of up to ±4% (Supplementary Figs. 2 and 3). The structural disorder of $MoS_2$ shows a random distribution of bonding lengths and angles caused by the deformation. The band gap fluctuations range up to ~1 eV (Fig. 1c) and show a strong correlation with the curvature (Supplementary Fig. 2j)[40]. Such a random distribution of the band gap can act as a random potential, which is analogous to the Anderson localization (see the Supplementary Information). The STS

maps at a fixed energy near the band edges (Fig. 1d, e) show the existence of localized states as isolated peaks decaying rapidly, while extended states over the space were observed at the energy above the conduction band edge (Fig. 1f, theoretical results are presented in Supplementary Fig. 4a–d). In contrast to the extended states, the localized states are not sensitive to the boundary conditions and do not contribute to the conductance at the 0 K limit[5]. In the STM/STS results, the curvature-induced effects were clearly distinguished from the defects and charged impurities.

### The flattening of band edges via doping
In Fig. 1g, j, m, local conduction band minima (CBM) and valence band maxima (VBM) were extracted from STS results (Supplementary Fig. 5b). At the neutral state, the fluctuation range of the VBM is similar to that of the CBM (Fig. 1g). The effect of the bending strain is clear at the curved region (green-shaded in Fig. 1h, i), where the CBM decreases while the VBM increases evenly, i.e., a band gap ($E_G$) reduction[40].

By electron-doping (Supplementary Fig. 1b), the spatial flattening of the CBM was observed (Fig. 1j). When additional electron charge is added by doping, the electron prefers the local minima of CBM (red-shaded in Fig. 1k, l) acting like a "charge basket" (see Supplementary Information for details). When the electron-doping charge is accumulated at the local minima of the CBM, the repulsive Coulomb interactions from the localized electrons increase the local chemical potential and shift the band edges up, resulting in local band bending (Fig. 1k, l). This local band bending increases until spatial equilibrium of the electrochemical potential or Fermi level ($E_F$) is achieved (the terminology is explained in the Supplementary Information). Eventually, the fluctuation of the CBM is reduced, i.e., spatial flattening of CBM (Fig. 1l).

For hole-doping, the spatial flattening of the VBM was observed (Fig. 1m). Similar to the electron-doping case, the hole is preferentially accumulated at the local maxima of the VBM until equilibrium of the electrochemical potential is achieved with downward local band bending due to the attractive Coulomb interactions by localized holes (Fig. 1n, o). Consequently, the curvature-induced band gap fluctuation acts like a charge basket, which leads to doping charge localization and an anomalous band edge flattening due to the Coulomb interactions by localized charge.

The doping-charge-induced band edge flattening of curved monolayer $MoS_2$ was confirmed by DFT calculations. Figure 2a shows the doping dependence of LDOS of curved monolayer $MoS_2$. The band gap is reduced in high curvature regions of the structure. The LDOS results clearly show that the curvature induces band gap fluctuations and the doping leads to the band edge flattening. The variation of band edges ($\Delta E$) in Fig. 2a is plotted as a function of doping concentration in Fig. 2b. At an electron-/hole-doping concentration of higher than $\sim 5 \times 10^{13}\,cm^{-2}$, the variation of band edges is saturated at a low value, indicating the spatial flattening of the band edge. The asymmetry of the flattening between conduction and valence band edges was induced by different orbital configurations and capacities of LDOS in the curvature regions (Supplementary Fig. 6).

Similar to the $MoS_2$ results, band edge flattening via doping was observed in various other TMdCs ($MX_2$, where M = {Mo, W} and X = {S, Se, Te}), which indicates that this is a general phenomenon in 2D semiconductors (Fig. 2c–h). Generally, 2D materials are flexible and their electronic structures are sensitive to the extrinsic factors such as a substrate and adsorbates. Moreover, 2D materials have strong Coulomb interactions due to poor dielectric screening by a small volume of themselves, which enhances the band edge flattening. Therefore, the curvature-induced band gap fluctuation and band edge flattening by doping are universal in 2D semiconductors. The degree of variations in band gap and band edge flattening depends on material properties.

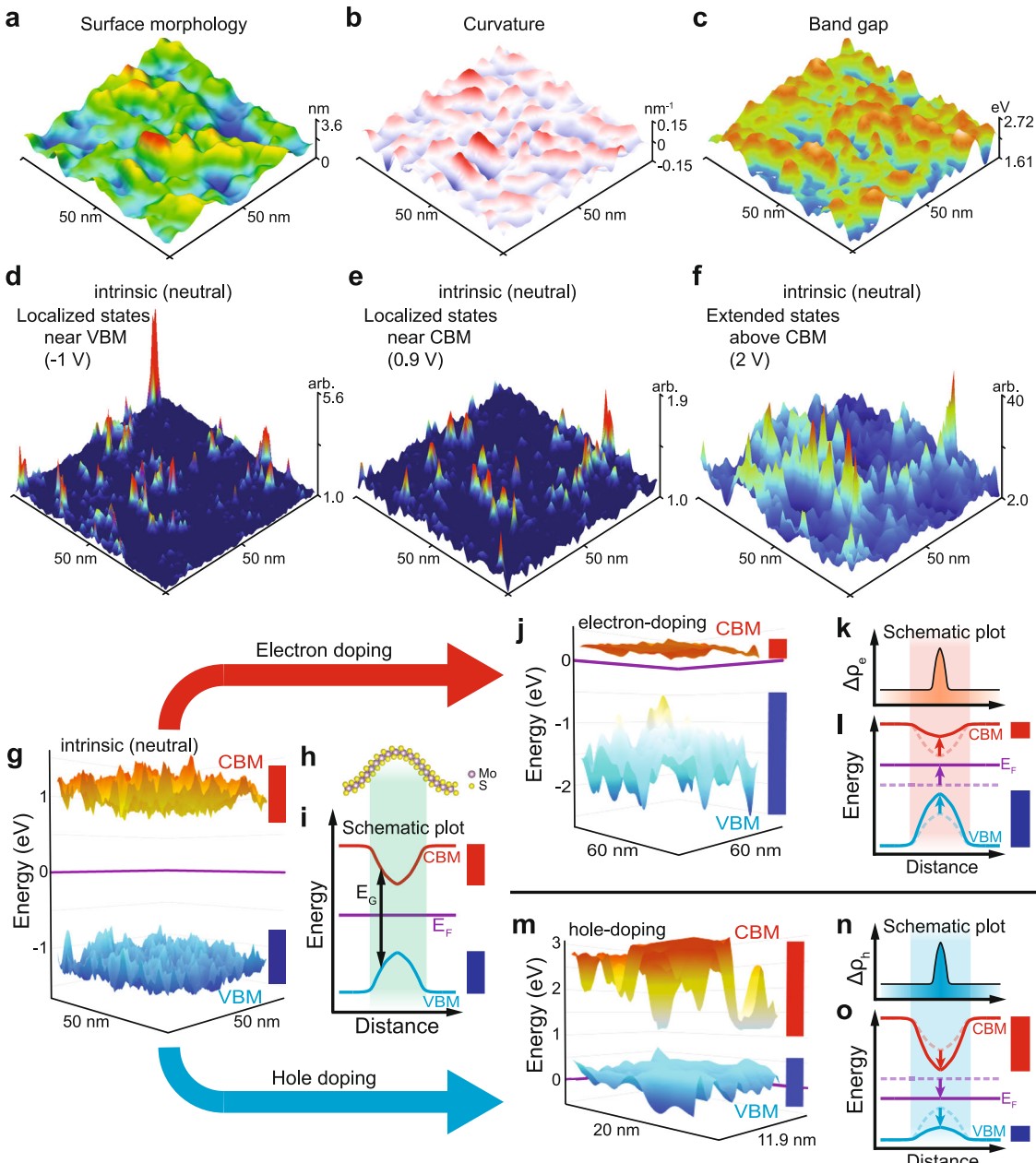

**Fig. 1 | Curvature-induced charge localization and band edge flattening in monolayer MoS2. a** A 3D topographic STM image. **b** Local curvatures and **c** curvature-induced band gap fluctuation in the area of (**a**). **d**–**f** STS maps for localized and extended states for the intrinsic state of disordered MoS₂. Localized peaks near (**d**) valence and (**e**) conduction band edges. **f** Extended states covered the whole measured surface. **g**–**i** Curvature-induced band edge fluctuation at a neutral state. **g** VBM and CBM in the area of (**a**). **h** Structural model of a single cylindrical curvature, and (**i**) its CBM, VBM, and band gap ($E_G$). **j**–**l** Spatial flattening of CBM via electron-doping. **j** Band edges at electron-doped state

($\Delta n_e = 5.67 \times 10^{12}$ cm$^{-2}$ (gate bias of 75 V)). Schematic plots of (**k**) doping charge localization and (**l**) its local band bending. **m**–**o** Spatial flattening of VBM via hole-doping. **m** Band edges at hole-doped state ($\Delta n_h = 4.54 \times 10^{12}$ cm$^{-2}$ (gate bias of −60 V)). ($\Delta n_{e(h)}$ denotes electron (hole) doping concentration). Schematic plots of (**n**) doping charge localization and (**o**) its local band bending. **k**, **n** $\Delta \rho_{e(h)}$ indicates electron (hole)-doping charge density. Purple lines indicate the Fermi level ($E_F$). In all cases, the lengths of red and blue side bars indicate fluctuation ranges of CBM and VBM, respectively. Dashed lines indicate before-doping.

## Variation of local work function with charge localization

To gain further insight into the curvature-induced charge localization, TBH maps were obtained using tip-height-dependent tunneling current spectroscopy (Fig. 3). In STM, the exponential dependence of the tunneling current on the tip-sample distance is strongly correlated with local work function (the required energy to withdraw an electron from the system)[41].

As shown in Fig. 3a, b, the surface morphology of monolayer MoS₂ (Fig. 3a) is correlated with its TBH map (Fig. 3b). For the area of Fig. 3a, the averaged line profiles of surface height (Fig. 3c), curvature

(Fig. 3d), and TBH (Fig. 3e) are plotted. The high curvature area has lower TBH (red-shaded) as expected since the localized doping electrons enhance the electron-electron repulsion at the high curvature area[41]. In the neutral state of MoS₂, the fluctuation range of the TBH becomes four times smaller (Supplementary Fig. 8) than that for electron-doped MoS₂ in Fig. 3b, due to its dilute charge carrier density.

In the calculated charge densities of the curved MoS₂ (Fig. 3f), the doping charge localization is clearly visible in the higher curvature regions (red-shaded in Fig. 3g). Moreover, the calculations show that

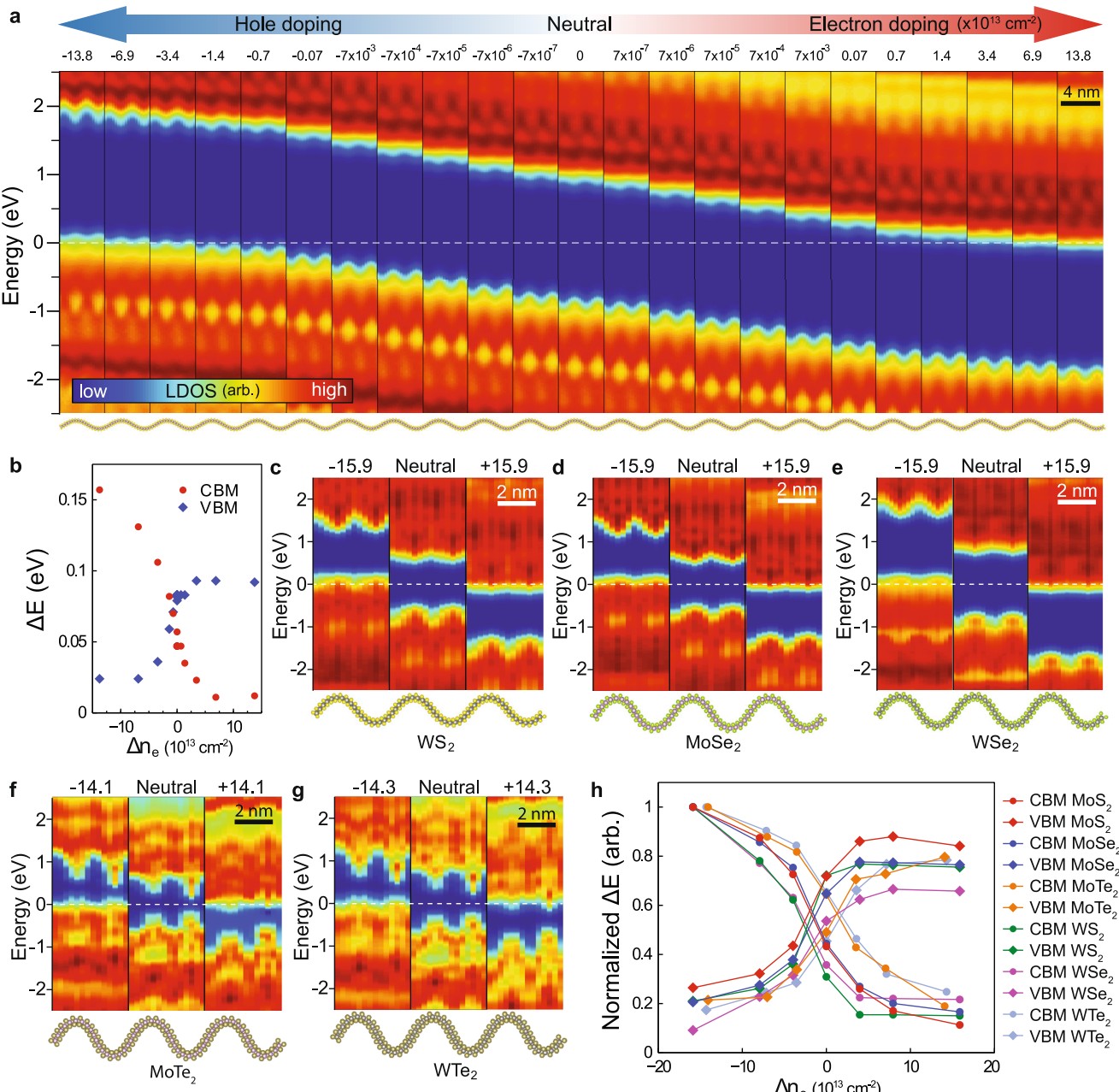

**Fig. 2 | DFT calculations showing a spatial flattening of band edges in various 2D semiconductors. a** The variation of the LDOS depending on the doping concentration of monolayer $MoS_2$ in cylindrical curvature structure. Corresponding atomic structure is shown under each column. Blue regions in LDOS indicate band gap. The Fermi level is set to zero (white-dashed line). Colored arrows at the top indicate the doping levels of each column. **b** Fluctuation range of band edges ($\Delta E$) as a function of doping concentration ($\Delta n_e$) extracted from the theoretical results in (**a**). **c–g** Doping induced spatial flattening of band edges in (**c**) $MoSe_2$, (**d**) $WSe_2$, (**e**) $WS_2$, (**f**) $MoTe_2$, and (**g**) $WTe_2$. Doping levels are indicated at the top of each column in the unit of $10^{13}$ cm$^{-2}$. All color scales for LDOS are the same in (**a**). **h** Fluctuation ranges of band edges as a function of the doping concentration of various 2D semiconducting monolayers, normalized by the maximum fluctuation range of CBM. In each plot, arb. denotes arbitrary units.

the range of local work function fluctuations relative to the mean value ($\Delta\Phi$) increases as the doping level is increased (Fig. 3h). For electron-doping, the local work function in the higher curvature region is minimized, in good agreement with the TBH result of Fig. 3e. For hole-doping, the local work function of the high curvature area increases due to the attractive Coulomb interaction by localized holes in the area, reversing the overall trend of fluctuation in local work function compared to the electron-doping cases. In addition, the fine features of local work function in the curvature regions for electron-doping cases are shifting from S to Mo atom sites as increasing the electron-doping level along with the flattening of both background and

fine features in the curvature regions as shown in Fig. 3h. The hole-doping cases did not show fine features and flattening of local work function in the curvature regions. The difference originated from orbital configurations near each band edge that differentiate the preference in occupation of states and capacity of the density of states relevant to the degree of the flattening of local work function by the charge screening in the curvature regions (Supplementary Figs. 6 and 9).

Other doped 2D semiconducting monolayers (MX$_2$, where M = {Mo, W} and X = {S, Se, Te}) also show curvature-induced charge localization (Supplementary Fig. 10).

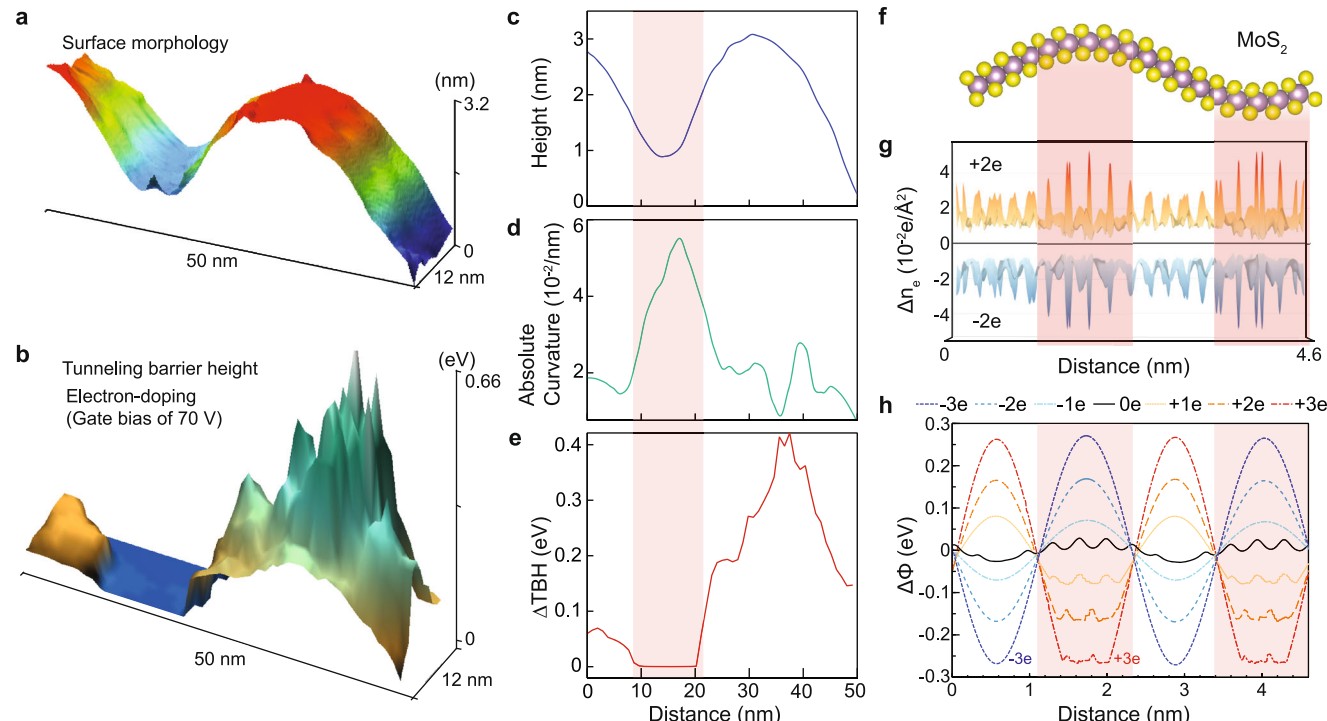

**Fig. 3 | Charge localization at a curved MoS2. a** STM topography and **b** TBH mapping at 70 V of gate bias as electron-doping. **c**–**e** the correlations among the averaged line profiles of (**c**) the surface morphology, (**d**) absolute curvature, and (**e**) relative TBH difference (*ΔTBH*) of the area shown in (**a**). The average values for plots were obtained in the axis of 12 nm in (**a**, **b**). (Supplementary Fig. 7 for each curvature value) The sample bias of −3 V was applied. **f** Structural model of curved monolayer MoS$_2$. **g** Calculated doping charge density ($\Delta n_e$) of ±2$e$/unit-cell in (**f**), where $e$ is the electron charge and the sign of +(−) is the addition of electrons (holes). **h** Calculated local work function variation ($\Delta \Phi$) in (**f**) depending on the doping concentration. Electron-/hole-doping of ±2$e$/unit-cell is ±15.8 × 10$^{13}$ cm$^{-2}$. High curvature regions are red-shaded.

## Criticality in the quantum fluctuation of wavefunctions

The substrate-induced random fluctuations of band edges act as a random potential for electrons or holes, leading to the static disorder that manifests MIT and band tail of localized states (Fig. 4).

The critical behaviors of MIT can be characterized by autocorrelation, multifractality, and a normalized distribution of LDOS extracted from STS results. The MIT in semiconducting TMdCs appears at the critical doping concentration, which can be achieved by controlled doping. The Fermi energy at the critical doping concentration is denoted critical energy ($E_C$).

Figure 4a–d shows LDOS maps at different energies, extracted from STS results of the electron-doped MoS$_2$ in Fig. 1j. The localized states are observed near the VBM (Fig. 4a) and CBM (Fig. 4b). In comparison with the localized states near the VBM, the localized states near the CBM are broader with lower intensity, owing to the shallow fluctuation of the CBM due to the band edge flattening via doping. In Fig. 4c, the criticality is observed at the critical energy which is determined by autocorrelation as shown in Fig. 4e (see below). The strong fluctuations of LDOS at the critical energy were exhibited over the entire area, which is predicted by theoretical results[1–4,33]. The extended states above the critical energy (Fig. 4d) show a uniform intensity regardless of the structural disorder, as a result of the band edge flattening by the doping charge. The crossover between localized and extended states confirms the MIT with the critical point, which exhibits a quantum phase transition from inhomogeneous insulating to homogeneous metallic states.

In Fig. 4e, the radial-averaged autocorrelation of the LDOS at an energy between two spatial points (data from Fig. 1j) can be divided into two different areas, a rapid decay (blue-colored) area and an almost constant (red-colored) area, by the critical energy[35]. In the radial-averaged autocorrelation results, the rapid decay into a lower value indicates localized states and the slow decay with a higher value corresponds to the extended states. The critical behavior obeys the power law, $\sim |\boldsymbol{R}|^{-\eta}$ where $\eta$ is an exponent and $|\boldsymbol{R}|$ is the distance between the two spatial points[3] (See the Methods). This critical behavior is clearly visible in line profiles (Fig. 4f) of autocorrelation (Fig. 4e) at fixed energies (linear in the log-log plot). For example, the line profiles at 0.395, 0.5, and 0.635 eV follow the power law indicating criticality[1–4]. The obtained $\eta$ at the critical energy of 0.635 eV is $-2.23 \times 10^{-3}$. The anomalous exponent of the second moment ($\Delta_2$) is extracted from the multifractality of the STS results and it is $-2.28 \times 10^{-3}$. Our results are surprisingly well-matched to the theory of multifractality in disordered systems[3], $\eta = -\Delta_2$ for the criticality. The obtained value of $\eta$, however, could not be matched with any previous theoretical expectations, which implies the necessity of a new theoretical approach with the band edge flattening. The autocorrelation shows the contour of $-(E_C - E)^{-\nu}$ with the critical exponent $\nu$ for $E < E_C$ (dashed line in Fig. 4e) and the divergent contour ($|\boldsymbol{R}|$ - infinity) for $E > E_C$ due to the very slow decay of the autocorrelation that is almost constant (the red-colored region in Fig. 4e), which is extracted from uniform LDOS maps (Fig. 4d).

In Fig. 4g, the singularity spectra ($f(\alpha)$) of the LDOS maps (Fig. 1j) at different energies confirm the multifractality of LDOS near or at the critical energy, where the peak position ($\alpha_0$) is larger than 2 and the width of the peak is broadened. Above the critical energy, the singularity spectrum approaches the metallic limit, where $f(\alpha)$ is only concentrated at $\alpha = 2$ with zero-width and $f(\alpha = 2) = 2$ ($f(\alpha) = -\infty$, otherwise), denoted by the vertical dotted line[1], and coincides with the autocorrelation results for extended states above the critical energy (Fig. 4d).

In Fig. 4h, the distribution of the normalized LDOS changes from log-normal to normal distributions, which corresponds to a transition

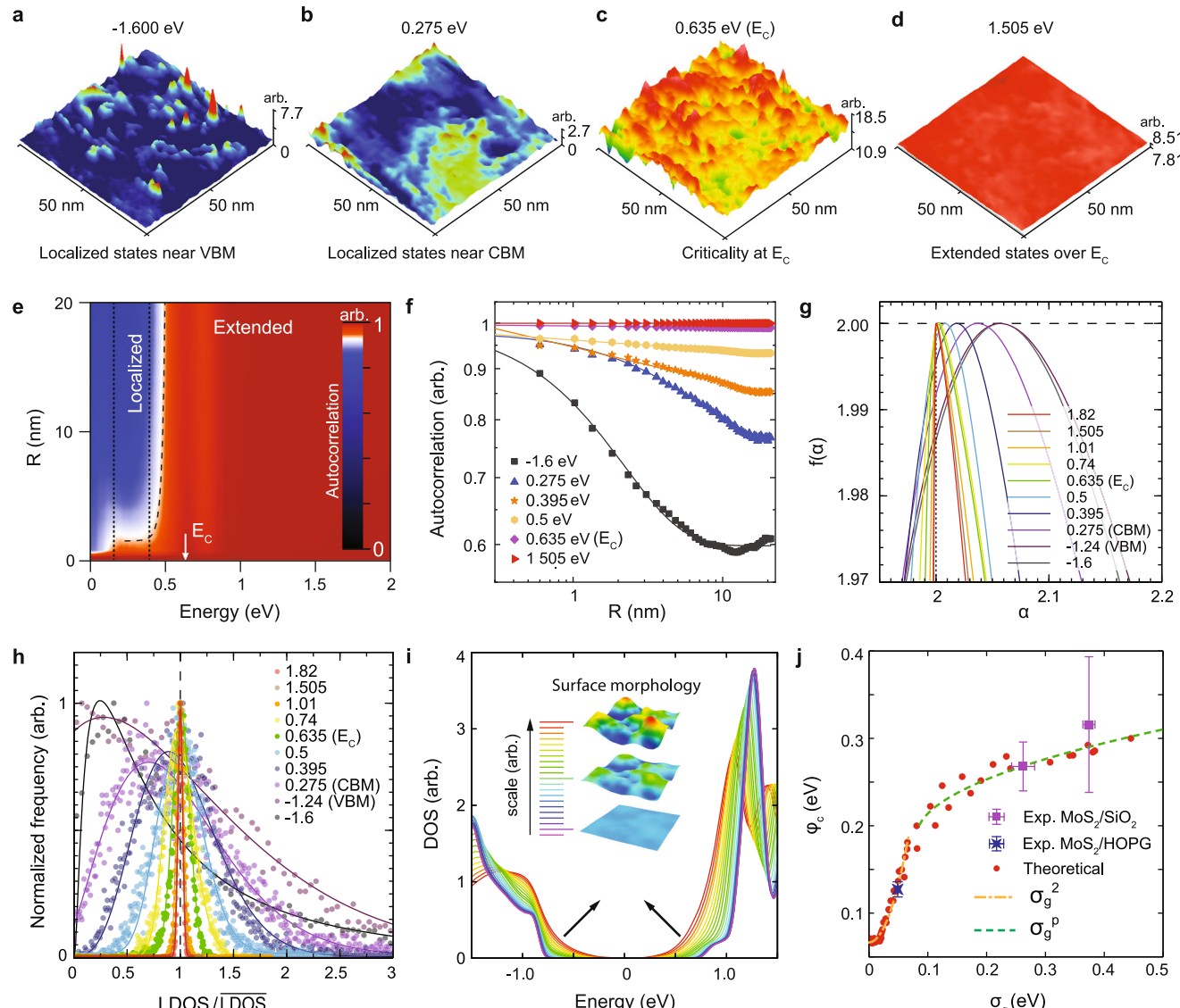

**Fig. 4 | Criticality of metal–insulator transition and band tails in structurally disordered monolayer MoS2. a–d** STS maps of the electron-doped MoS$_2$ (data from Fig. 1j) near (**a**) VBM, (**b**) CBM, (**c**) at the critical energy ($E_C$), and (**d**) above the critical energy. **e** Autocorrelation of the STS results in the electron-doped MoS$_2$ (data from Fig. 1j). The interval between the dotted lines indicates the fluctuation range of the local conduction band edges. The critical energy ($E_C$) is indicated by the arrow. The dashed line is - $(E_C - E)^{-\nu}$, following the contour of the autocorrelation. The Fermi level is set to zero. **f**, Line profiles of the autocorrelation at different energies. The colored lines are fitted lines using the power law or exponential function. **g** Multifractal spectra and **h**, histograms of normalized LDOS at different energies. **i** Exponential band tails near the band edges, calculated by the tight-binding method. Inset shows scaled structural models corresponding to the strength of the disorder. Arrows indicate the protrusion of the band tails. **j** Correlation between the standard deviation of band gap distribution ($\sigma_g$) and band tail width ($\varphi_c$). The dash-dotted and dashed lines were fitting results by $\sigma_g^2$ and $\sigma_g^p$ (p is a fitting parameter), respectively. The error bars indicate the standard deviations of data. The theoretical results were collected from the various surface morphologies with different roughness scaling. The experimental results of MoS$_2$ on HOPG (height fluctuation -±0.1 nm) and SiO$_2$ (-±1 nm) are plotted. Each data point in (**j**) was obtained out of the several data sets. And each of the data sets includes over -10$^4$ spectra. For each plot, arb. denotes arbitrary units.

from localized to extended states[33,34], in agreement with the autocorrelation and multifractality results. The sharpness of both singularity spectra near the metallic limit and normal distributions of the LDOS maps above the critical energy reflects the effect of doping-induced band edge flattening that leads to a uniform LDOS map (Fig. 4d) as explained by the autocorrelation results (Fig. 4e). Our results emphasize the importance of charge localization and band edge flattening in MIT. In addition, the band edge flattening induced by local chemical potential fluctuations is compatible with the disordered Fermi liquid picture including the screened (renormalized) potential[42].

The experimental results for the hole-doping case (Fig. 1m) in autocorrelation, multifractality, and normalized distribution of STS

agree well with the results of the electron-doping case (Supplementary Fig. 11).

## Band tail formation in structural disorder

The structural disorder is expected to form exponential band tails near the band edges in the density of states (DOS). The band tails are localized states, following -exp(−|E|/$\varphi_c$) where $\varphi_c$ is the characteristic width of the band tail[20,21]. In Fig. 4i, the formation of the exponential band tails in the DOS was confirmed in randomly deformed monolayer MoS$_2$, using tight-binding methods[43] with various scales of height fluctuations in surface morphology as shown in the inset with color scales. As the surface fluctuations in MoS$_2$ increase, exponential band tails near the

band edges protrude further (log-scale plots are in Supplementary Fig. 13a and experimental results are in Supplementary Fig. 12). Moreover, the experimentally observed LDOS maps (Fig. 1d–f) are reproduced in calculated LDOS maps (Supplementary Fig. 4a–d) and demonstrated localized and extended states. The localized and extended states with exponential band tails (Supplementary Fig. 13b, c) were reproduced again with another theoretical approach using the Schrödinger equation for random potential fluctuations, which assures the general behavior in Gaussian random potential fluctuations[20,21].

Figure 4j shows the relation between the characteristic width of the band tail and the standard deviation of band gap fluctuations ($\sigma_g$), which were extracted from LDOS results of several different surface morphologies. In the small or medium scale of surface fluctuations, $\varphi_c$ is proportional to $\sigma_g^2$, which is in good agreement with theoretical predictions of exponential band tails in a static disorder[20,21]. For large surface fluctuations, we found that $\varphi_c$ is proportional to $\sigma_g^p$ with $p = 0.11$ showing saturation behavior. Experimental results of band tails in various LDOS maps extracted from MoS$_2$ on highly oriented pyrolytic graphite (HOPG) and SiO$_2$ were consistent with theoretical results (Fig. 4j). The fluctuation of band edges (or band gap) by surface deformation follows a Gaussian (normal) probability distribution (Supplementary Fig. 14) that is only expected to produce an exponential band tail[20]. Therefore, the existence of exponential tails in the DOS near the band edges confirms the "random" potential under the central limit theorem. The obtained band tail widths for different disorder strengths (Fig. 4i) confirm the previously reported values of ~0.1 eV from transport experiments[30]. The band tails with localized states caused by structural disorder explain why charge trap states near band edges are dominant in monolayer MoS$_2$ on SiO$_2$ (refs. 30, 39). It is also compatible with a percolation picture for low doping density and percolation-induced MIT by thermal activation[39]. In addition, the structural-disorder-driven charge localization explains why the charge trap near band edges and critical doping charge density became smaller as increasing the thickness of MoS$_2$ (multilayers) in the previous results[39]. As increasing the thickness, the flexibility of MoS$_2$ decreased leading to smaller curvature formations allowing less capacity for charge localization (Supplementary Fig. 3).

## Symmetry classes and structural-disorder-driven magnetism

In the structural disorder, the curvature of monolayer MoS$_2$ is expected to change the local spin–orbit coupling (SOC) and induce a pseudomagnetic field as an emergent gauge[44,45]. The local variations in SOC should be investigated in a viewpoint of symmetry classes. The random change of SOC implies the symplectic class (time-reversal and broken spin-rotation symmetries) that allows 2D-MIT[46]. On the other hand, the previous theoretical expectation shows that the pseudomagnetic field in monolayer MoS$_2$ is proportional to the local Gaussian curvature[45], which breaks the time-reversal symmetry that might be contrary to 2D-MIT as in the unitary class[1,47]. The other theoretical result, however, confirmed that a weak magnetic field below the critical magnetic field with a random SOC still allows the 2D-MIT with enhancement of the $\nu$ exponent[47]. From the STM result in the same area of Fig. 1j, the average value of the Gaussian curvature ($\langle\kappa\rangle$) is ~5.36 × 10$^{-7}$ Å$^{-2}$ and corresponds to the spatial average value of the pseudomagnetic field ($\langle B\rangle$) of ~0.021 T. The value of the pseudomagnetic field was evaluated by $\langle B\rangle = (\hbar/2e)\langle\kappa\rangle$ where $e$ is the electron charge and $\hbar$ is the Planck constant divided by 2π (ref. 45). From the ref. 47, the equation of the $\nu$ as a function of the magnetic field gives ~7.19 with $\langle B\rangle = $ ~0.021 T (Supplementary Fig. 15), which is in good agreement with the fitting result of ~7.28, $\nu$ from ~$(E_C − E)^{-\nu}$ by fitting of the contour in Fig. 4e (For the detail, see the Supplementary Information). The tail of the fitted curve (~$(E_C − E)^{-\nu}$, dashed line in Fig. 4e) was saturated at 1.54 nm (~5 lattice constants) below $E_C$, implying the upper limit of localization length in a fully localized monolayer MoS$_2$ (ref. 32). On the other hand, the fitting value of $\nu$ obtained from the localization/correlation lengths

characterized in the radial-averaged autocorrelation profile was 2.73 (Supplementary Fig. 16). All the results of $\nu$ from independent approaches coincided with each other. Those results imply that the MIT in monolayer MoS$_2$ is not clearly predicted by its symmetry (for example, time-reversal and spin rotation symmetries) under the structural disorder.

To confirm the pseudomagnetic field effect, we calculated magnetization in a spherical curvature as a motif using noncollinear DFT. When it is doped with electrons, the highest magnitude of magnetic moments was located at the center of the spherical curvature (Fig. 5a), correlated with the localized doping charge (Fig. 5b). For hole-doping, see Supplementary Fig. 17. It is noteworthy that the doping charge is predominantly localized at the Mo sites and localization of charge appears at the center of the curvature (Figs. 3 and 5b).

To investigate the disorder effect in local magnetic moments, the structural model of MoS$_2$ with random surface fluctuation (area of 9.48 × 8.76 nm$^2$) was calculated using DFT (Fig. 5c). The correlation between local curvatures and localization of charge near the band edges in the structural model was confirmed again (Supplementary Fig. 4e–h).

Figure 5d, e shows a map of the magnitude of the local magnetization (Fig. 5d) and $z$-axis-emphasized 3D vector plot (Fig. 5e) of the structure in Fig. 5c at a doping level of 66 $e$/unit-cell. The calculated intrinsic magnetization in the neutral case of Fig. 5c is ~2.0 × 10$^{-7}$ $\mu_B$/Å$^3$, and a pseudomagnetic field induced by Gaussian curvature in the STM result (Fig. 1j) produces the magnetization of ~4.0 × 10$^{-7}$ $\mu_B$/Å$^3$ in good agreement with the previous experimental result of ~0.004 emu/cm$^3$ (~4.3 × 10$^{-7}$ $\mu_B$/Å$^3$) in monolayer MoS$_2$ on SiO$_2$ (refs. 36, 38, 45). Furthermore, the calculated magnetoconductance in the curvature-induced pseudomagnetic field agrees well with the previous experimental results of the negative magnetoconductance in monolayer MoS$_2$ (ref. 45). Noted that, the DFT-calculated magnetization was in the range of 10$^{-8}$ to 10$^{-4}$ $\mu_B$/Å$^3$ depending on various structures and doping levels in MoS$_2$, where $\mu_B$ is the Bohr magneton. Interestingly, the arrangement of magnetic moments under the structural disorder exhibited non-uniform ferromagnetic domains (Fig. 5d, e). The order of magnetization was purely governed by the strength of the structural disorder determined by the degree of geometrical deformation. These results suggest that the randomly deformed 2D semiconducting TMdC monolayers are a new class of spin glass systems in which topological and magnetic disorders coexist. Direct measurements of local magnetization in atomic scale are needed to understand the criticality of quantum fluctuations further, using spin-polarized STM or magnetic force microscope.

In summary, there are three major effects of structural disorder in semiconducting TMdCs. First, the curvature-induced band gap (edge) fluctuations act like charge baskets leading to the localization of charge in a random potential, which explains the MIT associated with band edge flattening towards uniform metallic states. Second, the localized states in the structural disorder formed the exponential band tails near the band edges, confirmed experimentally and theoretically. Last, magnetic moments emerge due to the curvature-induced change of the spin–orbit interaction, which correlates with doping charge localization. These results help to elucidate the electronic, magnetic, and transport properties of structurally disordered flexible 2D semiconductors towards quantum applications.

## Methods

### Synthesis of MoS$_2$

Monolayer MoS$_2$ films were grown under low pressure by metal-organic chemical vapor deposition (MOCVD)[48]. Molybdenum hexacarbonyl (Mo(CO)$_6$, Sigma Aldrich) and diethyl sulfide ((C$_2$H$_5$)$_2$S, Sigma Aldrich) which were selected as precursors of Mo and S, respectively, were supplied in the gas phase into a one-inch quartz tube furnace using a bubbler system with Ar as the carrier gas.

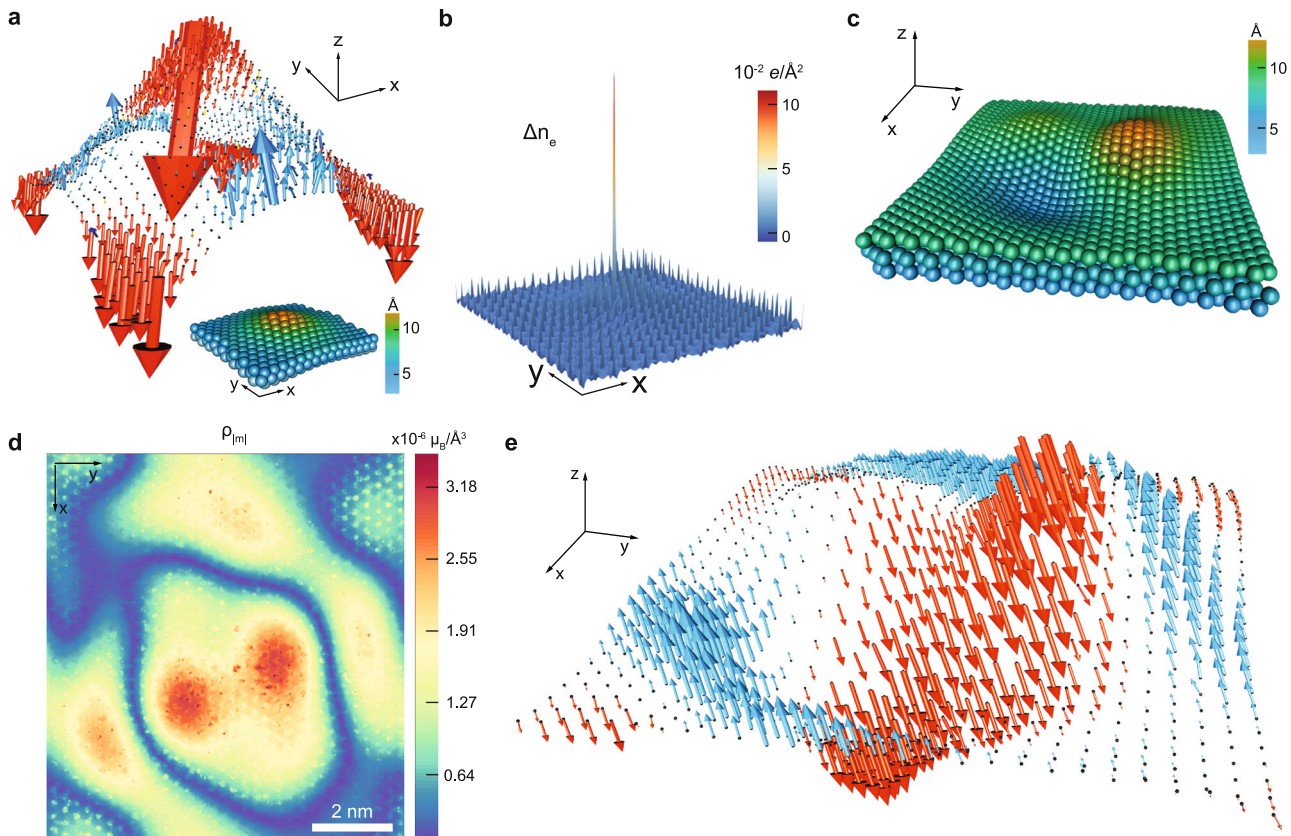

**Fig. 5 | Curvature-induced magnetism in monolayer MoS$_2$. a** DFT-calculation of local magnetic moments in a spherical curvature structure of doped MoS$_2$ (+4$e$/unit-cell). The surface of the structure is indicated by dots with the magnified $z$-axis for better visualization. Inset at the bottom is the atomic structure (4.74 × 4.38 nm$^2$). **b** The calculated electron-doping charge density ($\Delta\rho_e$) in the structure of (**a**). **c** A randomly deformed structural model of MoS$_2$ (9.48 × 8.76 nm$^2$) with the magnified $z$-axis. **d** DFT-calculated magnitude of local magnetization ($\rho_{|m|}$) and **e** local magnetic moments in (**c**) with electron-doping (+66$e$/unit-cell), exhibiting localized non-uniform magnetic domains. The dots in (**e**) indicate the surface of (**c**) with the magnified $z$-axis.

The MoS$_2$ film was synthesized on a 300 nm-thick SiO$_2$ layer on a Si wafer with a flow rate of 100 sccm for Ar, 0.1 sccm for Molybdenum hexacarbonyl, and 1.0 sccm for diethyl sulfide at a growth temperature of less than 350 °C. The growth time was about 20 h. After growth, the furnace temperature was ramped down to room temperature. The quality of the monolayer MoS$_2$ films was characterized by Raman spectroscopy and photoluminescence (Supplementary Fig. 1c, d).

**Transfer of MoS$_2$**
Poly(methyl methacrylate) (PMMA) was spun onto MOCVD-grown MoS$_2$ on SiO$_2$ at 1000 rpm. The PMMA-coated sample was slowly dipped into a 2 M KOH solution. Subsequently, the detached PMMA/ MoS$_2$ film from the SiO$_2$ was suspended in distilled water to remove the remaining residues. After transfer of the film onto a target substrate, the sample was soaked in acetone and isopropyl alcohol baths to remove the PMMA. Finally, electrodes were fabricated by metal deposition.

**Gate-tunable scanning tunneling microscopy and spectroscopy**
The scanning tunneling microscopy (STM) and spectroscopy (STS) were performed using a home-built STM with a gate-tunable configuration at ~4.8 K under ultra-high vacuum (UHV) of -10$^{-10}$ mbar. Chemically etched W tips were used for the STM probe. For STS and tunneling barrier height (TBH) spectroscopy, conventional lock-in techniques were applied with an oscillation frequency of 417 Hz and amplitude of 7 mV (5 pm for TBH). An apparent barrier height (TBH) is defined as TBH = $\frac{\hbar^2}{8m_e}\left(\frac{d\ln I}{ds}\right)^2$ where $s$ (in Å) is the distance between

sample and tip, $m_e$ is the electron mass, $\hbar$ is the reduced Planck constant, and the unit of TBH is given in eV. The $\Delta TBH$ is defined by the difference between TBH and minimum of TBH (i.e., $\Delta TBH$ = TBH − minimum of TBH). The TBH was taken at the distance of ~6 Å between sample and tip (i.e., $s$ = ~6 Å).

All samples were outgassed below 350 °C in UHV. For doping, a back-gate bias was applied through an arsenic-doped Si substrate coated with a 285 nm-thick SiO$_2$ layer. Palladium electrodes were used for gating and biasing of samples.

**Strain map, curvature, and band gap analysis**
The STM images of MoS$_2$ in atomic-scale resolution show the sulfur-induced protrusions with the honeycomb lattice symmetry (Supplementary Fig. 1), which corresponds to the isosurface of charge density. A local lattice parameter along the deformation of monolayer MoS$_2$ on a substrate can be obtained by fast Fourier transform (FFT) analysis of each local domain of ~2.3 × 2.3 nm$^2$ over the measured surface. The local curvature in an STM image should be evaluated over a few nanometers (overall shape of height fluctuation), in which small corrugation of charge density among the sulfur-induced protrusions in the ångström-scale was not considered because the structural curvature was developed by the positions of atoms, not the orbital shape of each atom (Supplementary Fig. 5a). The local curvature was measured by the mean curvature or local maximum of principal curvatures, which showed high correlation with band gap (Supplementary Fig. 2). The mean curvature and local maximum of principal curvatures exhibited nearly the same cross-correlation with band gap.

## Characterization of exponential band tails

Exponential band tail width ($\varphi_c$) was extracted from STS mapping results as an average value of each band tail width from valence and conduction band edges. Band gap fluctuation was measured by the variance of band gap fluctuation ($\sigma_g$) as a representative value of the structural disorder strength. Under the assumption of the rigid shift in the local band bending, fluctuation of band gap is invariant in the single particle limit that is a representative measure of band edge fluctuations at any doping level with band edge flattening.

## Multifractal spectrum of energy-resolved STS map

The singularity spectrum ($f(\alpha)$) from STS mapping results was obtained based on the concept of the generalized inverse participation ratio,

$$P_q = \int d^d \boldsymbol{r} |\psi(\boldsymbol{r})|^{2q}, \langle P_q \rangle \propto L^{-\tau_q} \tag{1}$$

where $\psi(\boldsymbol{r})$ is a normalized electronic wavefunction, $L$ is the linear dimension of a $d$-dimensional system, and $\tau_q$ is an exponent related to generalized fractal dimension, satisfying $\tau_q = d(q-1) + \Delta_q$, where $\Delta_q$ are anomalous multifractal exponents[1-4]. In a discretized $d$-dimensional system with volume $L^d$ and boxes of linear size $l_b$, the generalized inverse participation ratio can be defined by

$$P_i^q(l_b) = \sum_{j \in \Omega_i(l_b)} |\psi(\boldsymbol{r}_j)|^{2q} \tag{2}$$

where $P_i$ ($q = 1$) is the probability of finding an electron in the $i$-th box ($\Omega_i$). A singularity strength $\alpha$ is defined by $P_i \propto \lambda^{\alpha_i}$, where $\lambda \equiv l_b/L$. If the number of boxes $N(\alpha)$ is counted for that $P_i \propto \lambda^{\alpha_i}$ where $\alpha_i$ is in between $\alpha$ and $\alpha + d\alpha$, then $f(\alpha)$ can be introduced by $N(\alpha) \propto \lambda^{-f(\alpha)}$, which indicates the fractal dimension of the points where the $N(\alpha)$ boxes are counted. The singularity spectrum $f(\alpha)$ is defined from the $\tau_q$ exponents via a Legendre transformation, $\tau_q = q\alpha - f(\alpha)$, $q = f'(\alpha)$, $\alpha = \tau_q'$. The singularity spectrum $f(q)$ and average of the singularity strength $\alpha(q)$ are alternatively defined by

$$f(\alpha(q)) = \lim_{\lambda \to 0} \frac{1}{\ln \lambda} \sum_i^N \mu_i(q, l_b) \ln \mu_i(q, l_b) \tag{3}$$

$$\alpha(q) = \lim_{\lambda \to 0} \frac{1}{\ln \lambda} \sum_i^N \mu_i(q, l_b) \ln P_i(l_b) \tag{4}$$

where $\mu_i(q, l_b) = P_i^q / \sum P_j^q$ and $N = \lambda^{-d}$ (ref. [49]). The $f(\alpha(q))$ and $\alpha(q)$ were used for STS analysis. A value of STS (i.e., $dI/dV$) is proportional to $|\psi_E(\boldsymbol{r}_i)|^2$ at a given energy $E$ as a probability of finding an electron at a position of $\boldsymbol{r}_i$, which is corresponding to the local density of states (LDOS), under the Tersoff–Hamann approximation[50].

The normalized distribution of the STS mapping results by the average value of STS results (LDOS/$\overline{\text{LDOS}}$ in Fig. 4h) exhibits a Gaussian distribution for extended states and a log-normal distribution for localized states. Localized states exhibit peaks in an LDOS map which show large deviations from the mean value leading to a log-normal distribution. On the other, extended states show smaller deviations from the mean value, following a Gaussian (normal) distribution. The transition from Gaussian to log-normal distributions characterizes the metal–insulator transition with the critical behavior of scaling anomaly in the singularity spectrum $f(\alpha)$ (refs. [33], [34]).

## Two-dimensional auto- and cross-correlation calculations

The two-dimensional auto- and cross-correlation were calculated for the evaluation of the spatial correlation of a quantity, resemblance, or correlation between two different image results. The cross-correlation were calculated by $S_C(x,y) = \sum_{k=0}^{M-1} \sum_{l=0}^{N-1-y} I(k,l) \times J(k+x, l+y)$, where $I$

and $J$ are images of $M \times N$ pixels along with coordinates of $(x,y)$. The autocorrelation is a kind of cross-correlation of the single image $I$ itself, calculated by $S_A(x,y) = \sum_{k=0}^{M-1} \sum_{l=0}^{N-1-y} I(k,l) \times I(k+x, l+y)$. The autocorrelation for the non-periodic images can measure the intrinsic resemblance of an image itself with a maximum peak at the center. For a single-valued image ($I(x,y) = $ constant), autocorrelation results in a constant value. In the cross-correlation, the strong highest peak at the center indicates a strong correlation between two different images.

The radial-averaged autocorrelation $C(E, \boldsymbol{R})$ was calculated to investigate the localization or correlation length near the critical point, which shows a relation with distance $|\boldsymbol{R}|$ between two points at the energy of $E$,

$$C(E, \boldsymbol{R}) = \frac{1}{2\pi} \int d\theta \int d^2\boldsymbol{r} g(E, \boldsymbol{r}) g(E, \boldsymbol{r} + \boldsymbol{R}) \tag{5}$$

where $g(E, \boldsymbol{r})$ is the local differential tunneling conductance ($dI/dV$) that is proportional to LDOS ($\rho(E, \boldsymbol{r})$) at the energy $E$. The radial-averaged autocorrelation was normalized by $C(E, \boldsymbol{0})$. The autocorrelation represented by LDOS (cf. Eq. (5)) near the critical energy shows a scaling law,

$$\langle \rho(E, \boldsymbol{r}) \rho(E, \boldsymbol{r} + \boldsymbol{R}) \rangle / \langle \rho(E) \rangle^2 \propto (\mathcal{L}/|\boldsymbol{R}|)^\eta, l < |\boldsymbol{R}| < \mathcal{L} \tag{6}$$

where the exponent $\eta = d - \tau_2 \equiv -\Delta_2$, and $\langle \rho(E) \rangle$ is the disorder-averaged LDOS at the energy $E$ that is counted from the electrochemical potential (Fermi level), $l$ is the length scale of elastic scattering mean free path, $\mathcal{L} = \min\{\xi, L_\phi, L\}$ is the shortest length among the localization or correlation length $\xi$, dephasing length $L_\phi$, and system size $L$ (refs. [1], [3]). If the LDOS map is uniform and almost single-valued over the space as an extended state, the autocorrelation is almost constant with very slow decay (Fig. 4d–f). For the localized states, the isolated peaks in the LDOS map show the decaying behaviors in the autocorrelation (Fig. 4a, b, e, f).

## Density functional theory calculations

The first-principles calculations were performed to investigate the electronic structures of curvatures in the various transition metal dichalcogenides (TMdCs) monolayers (MX$_2$, M = {Mo, W}, X = {S, Se, Te}) using the Vienna ab initio simulation package (VASP) based on the density functional theory (DFT) with a plane-wave basis set[51]. The pseudopotentials in the projector augmented wave (PAW) formalism with the Perdew–Burke–Ernzerhof (PBE) parametrization of the general gradient approximation (GGA) were used as implemented in VASP[52,53]. The cylindrical and spherical curvature structures of TMdCs were constructed in a slab geometry with a vacuum of -14 Å or more to avoid artificial effects, and all structural models were relaxed until the residual forces of each atom were less than 0.005 eV/Å. The plane-wave cut-off energy was 400 eV. For the surface Brillouin zone integration of cylindrical curvature structures, a 2 × 36 grid in the Monkhorst–Pack special $k$-point scheme with Γ-point was used as implemented in VASP[54]. For spherical curvature structures and larger unit cells, Γ-point was used due to a heavy computational load. The randomly deformed structures in a wide range of 94.80 × 87.57 Å$^2$ were fully relaxed until the residual forces of each atom were less than 0.04 eV/Å with constraints of surface height of few atoms to mimic the substrate-induced deformation of the structure under the elastic limit[55]. In all calculations, energy convergence was achieved with a tolerance of $10^{-6}$ eV. Non-collinear DFT calculations including spin–orbit interactions were performed as implanted in VASP based on the generalized local-spin density theory[56]. Several easy axes for orientations of local magnetic moments were investigated with orthorhombic unit cells. The work functions were obtained from calculations of electrostatic potentials without the exchange-correlation part.

**Tight-binding method and the Schrödinger equation approach**

For a wide structural model ($25 \times 25$ nm$^2$), the tight-binding methods with the three-band model of TMdCs were applied to calculate the random fluctuation of local surface height in monolayer MoS$_2$ (ref. [44]). For a set of scaling calculations, the disorder strength was scaled by control of surface fluctuation range from flat (0 nm) to ~4 nm. The many different surface morphologies were calculated to obtain the relation between the characteristic band tail width ($\varphi_c$) and standard deviation of band gap distribution ($\sigma_g$). The three-band model of TMdCs includes $d$-orbital ($d_{z^2}$, $d_{xy}$, and $d_{x^2-y^2}$) which is dominant near conduction and valence band edges[44]. The randomly deformed structural models within the third-nearest-neighbor were calculated to obtain LDOS of each orbital. The local fluctuations of surface height were generated randomly by Gaussian functions with the criterion that a ratio of height fluctuation to deformed area $\ll 1$ nm$^{-1}$ and periodic boundary condition was applied to avoid a boundary effect. The changes of hopping parameters $t_{ij}$ between $i$ and $j$ sites by strain were treated as

$$t_{ij} = t_{ij}^0 \exp\left(-\beta\left(\frac{a_{ij}}{a_0} - 1\right)\right) \qquad (7)$$

where $a_0$, $t_{ij}^0$ and $a_{ij}$ are unstrained lattice distance, primitive hopping parameter, and strained distance between $i$ and $j$ sites, respectively. The factor $\beta$ was chosen as 5 by the empirical orbital dependence[57]. The inclusion of spin–orbit interaction did not violate the conclusion for the curvature-induced charge localization mechanism.

Using the Schrödinger equation, the eigenstates of a random potential fluctuation over a honeycomb lattice potential were calculated. The calculations were performed by the time-independent Schrödinger equation with a random potential. The Hamiltonian operator $H$, $H = K + V_{random} + V_{lattice}$ where $K$ is the kinetic energy operator as the Laplace operator, $V_{random}$ is a random potential energy operator to mimic the curvature-induced random fluctuation of band edge for charge carriers, and $V_{lattice}$ is a periodic honeycomb potential energy operator corresponding to the potential of Mo atoms in MoS$_2$. The random potential was generated by a collection of random Gaussian functions. The maximum fluctuation range of the random potential was up to ~1 eV to mimic the experimental observation. From the observation in the experimental results, the sharpness of the random fluctuation was limited by a criterion such that spatial change of potential $\ll 1$ eV/nm. The Schrödinger equation approach shows formations of the exponential band tails and that the characteristic band tail width is proportional to the strength of structural disorder (Supplementary Fig. 13b).

## Data availability

The authors declare that all the data supporting the findings of this study are either shown in the main and supplementary text or available from the corresponding author upon request. Source data are provided with this paper.

## Code availability

The codes used for the analysis and calculations are available from the corresponding author upon request.

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

## Acknowledgements

We thank K. Kern for fruitful discussions and support. J.-H.P. and J.K. acknowledge the support from the U.S. Army Research Office (ARO) MURI project under grant number W911NF-18-1-0431 and the U.S. Army Research Office through the Institute for Soldier Nanotechnologies at MIT, under cooperative agreement number W911NF-18-2-0048.

## Author contributions

B.G.S. and S.J.J. designed and developed this work. B.G.S. conducted the STM/STS measurement and analyzed the STM/STS results. J.P. and J.K. performed the growth of the $MoS_2$ samples and characterized the samples. B.G.S. carried out DFT and tight-binding calculations. B.G.S. and Y.J. analyzed the strain and curvature in $MoS_2$. B.G.S. and S.J.J. wrote the manuscript. All authors participated in the manuscript review.

## Funding

## Competing interests

The authors declare no competing interests.
