## [Peer Review File · Nature Communications]

REVIEWER COMMENTS

Reviewer #1 (Remarks to the Author):

In this work, Bong Gyu Shin and co-workers studied the quantum fluctuations in disordered 2D semiconductors. By means of gate-tunable STM and STS measurements, the authors demonstrate that the curvature of MoS₂ localizes the charge to weaken the strain-induced band gap renormalization, leading to band edge flattening, which is further supported by theoretical calculations. The study of critical energy with the crossover between localized and extended states also offers great potential in understanding the metal-insulator transition in disordered 2D semiconductors. In addition, the authors explore the disorder effect in local magnetic moment due to the change of local spin orbit coupling. I recommend the publication of this work in Nature Communications. Before that, addressing the following issues can further strengthen the paper.

1. The authors need to specify the method/details on how to determine the energy position of the band edges with band tails? In addition, the experimentally observed STS in randomly deformed monolayer MoS shall be presented in parallel with the theoretical DOS using tight-binding methods (Figure 4i). In addition,
2. Some localized states might be linked to the surface morphology or curvature. Have the author considered other possible sources of localized states near band edges, such as sulfur vacancies or charge impurities on SiO₂. Can the authors estimate the density of defects and comments their impact on the electronic disorder?
3. The occurrence of MIT has not been confirmed experimentally because the gate injected carrier density does not reach the critical doping. In this context, some sentences (line 22-25) in the abstract and line 66-71 might be misleading.
4. In line 149, authors conclude that "the variation of band edges is saturated at a low value, indicating the spatial flattening of the band edge". However, it seems that the saturation is due to the heavily doping that the Fermi level has already touched the band edge rather than the band edge flattening. The authors should compare the band edge energy with MoS₂ in the same doping level but without curvature.
5. It is not very clear how the authors figure out the relationship between surface morphology and curvature as shown in Figure 3c, d. Should the hill and valley regions have the opposite sign of curvature? The authors need to explain this.
6. How to explain the fine features in Fig.3h? And why the electron and hole doping has an unsymmetric effect to the local work function variation (hole doping has fine features but electron doping does not) ?
7. In Fig. 4d and line 214, authors conclude that "the uniform intensity regardless of the structural disorder is a result of the band edge flattening", but in Fig.1f it shows that the extended states still show a big variation.

8. A discussion of the experimental realization for the curvature-induced magnetism in disordered 2D semiconductors shall be included.

Some minor issues to be addressed:

9. In Fig. 4j, are three data points sufficient to demonstrate the correlation between the standard deviation of the band gap distribution and band tail width?

10. Authors should mark the gate voltage for electron and hole doping in Fig. 1j and 1m, and estimate the charge carrier density based on the sample configuration.

11. The authors should specify the sample bias they applied for measuring the TBH.

Reviewer #2 (Remarks to the Author):

Compared with traditional semiconductors, 2D semiconductors are flexible and atomically thin and can adjust its surface profile to the landscape of substrates. This makes the metal-insulating transition (MIT) mechanisms in 2D semiconductors distinct from traditional semiconductors. Although many works have been conducted to study this effect in 2D materials, majority of them are from a macroscopic point of view, such as transport and capacitance, and the driving mechanisms are debating and not well understood. In this work entitled "Structural-disorder-driven critical quantum fluctuation and localization in two-dimensional semiconductors", Shin et al. investigated the metal-insulating transition mechanisms in 2D semiconductors from the microscopic point of view. Direct STM and STS are performed in a wide members of TMDC family on substrates, including MoS₂, WS₂, etc.. Random curvature in TMDS/SiO₂ induced band reconfiguration and localization states are observed near the band edge. Electron/hole Doping will flat the band edge of conductance/valence band and extended states will dominate. The observations can well explain the MIT in 2D materials and agree with the overall picture of percolation-transition. DFT calculations also well support the experimental observations. Overall, the topic about MIT is interesting and attracts researchers' attention. The experimental results, theory calculation and analysis are solid. I recommend for publication after addressing my concerns.

1. As shown in the introduction part "MoS₂ encapsulated in hBN does not show MIT, since hBN is very flat". Based on your proposed theory, what's the critical "defect density" (more precisely, the critical degree of curvature variations in MoS₂) that allows MIT occur in MoS₂?

2. Similar to the first question, based on my understanding of your proposed model, as long as there is random curvature in MoS₂, there will be localization states near band edge. Flatter surface will

result in narrower band tail and lead to smaller E_c . If it is true, MIT should be universal in MoS₂ but with variation of critical MIT transition energy/carrier density, since there is no perfect MoS₂ flakes. Is this correct?

3. Please discuss on the MIT transition in thicker samples based on your proposed theory and compare with previous experimental results (Nat Commun 6, 6088, 2015) if possible.

4. Other defects, such as vacancies, also exist in MoS₂. How to understand the “joint” effect of these defects and your proposed curvature-induced localization to MIT.

5. Previous STS/STM works on graphene have shown puddle-like structures. Could you please compare the difference between your theory and theirs.

6. Please show more physical meaning of R and “autocorrection” shown in Fig. 4, which is not clear enough to me.

7. Fig. 1i, k, n are theoretical calculated or schematic. Please put clearly in the legend.

8. Since the manuscript is relatively long, I suggest to put some subtitles to increase the readability of the manuscript.

Reviewer #3 (Remarks to the Author):

The authors studied the inhomogeneous electronic structure in a corrugated 2D semiconductor MoS₂ using gate-tunable scanning tunneling microscopy and spectroscopy. They have observed that the structure disorder results in the band gap's spatial variation, which can be flattened by electron(or hole)-doping, and the localized states forming band tails. In addition, the autocorrelation functions and histograms of the energy-dependent LDOS maps exhibit the power-law behavior and the log-normal distribution that have been theoretically expected as a hallmark for the quantum criticality of metal-insulator transitions. Although they do not provide a new perspective for the main point, their results will be constructive in understanding the electronic properties of flexible 2D semiconductors. However, given the informative results in this field, the following issues should be carefully addressed before I can recommend its publication in Nature Communications.

1) They used STM images to determine the curvature of the curved atomic structure. However, the contrast (z-height) of the STM image is determined by the atomic and electronic structure of the sample. A spatially inhomogeneous electronic structure also leads to contrast variation in the STM image. Please address how authors can exclude or minimize this effect.

2) They did not show full-range dI/dV spectra. It is necessary to offer a few representative dI/dV curves and explain how to determine the band gaps to construct the images shown in Fig. 1c.

3) Their DFT calculations showed the band gap variation determined by the curvature of the atomic structure for several semiconducting TMdC materials in Fig.2. Here, I cannot find the specific reason why it is necessary to include other materials. It would be better to focus on MoS₂ in the main text and move the results of other compounds to the supplementary.

4) Their DFT calculations show that CBM exhibits more pronounced spatial variations than VBM in the cylindrical curvature structure. The physical reason for the asymmetry needs to be better explained in the main text.

5) In Fig. 3, the authors show the spatial variation of the tunneling barrier height (TBH) and its correlation with surface morphology. However, many things need to be clarified in their plots and analysis. First, the unit of TBH must be eV, as shown in Fig. 3h. Second, how is the TBH surprisingly constant at the high curvature region? Third, the curvature plot in Fig. 3d must show the negative value on the hill like their extended data Fig. 3. Last, they should have described the details of how to measure TBH.

6) The authors analyzed the energy-dependent LDOS maps with autocorrelation functions and normalized histograms. Other people also used a similar way to prove the quantum criticality of metal-insulator transitions with the power-law behavior (Fig. 4e and f) and the log-normal distribution (Fig. 4g and h). To make their main idea more convincing, I strongly recommend showing the results of the "neutral" and "hole-doped" MoS₂. If they saw the differences or similarities, it would be great to explain them.

7) The authors argued that the randomly curved atomic structure (shown in STM images) leads to band tails. And their tight-binding calculations display the DOS spectra in Fig. 4i. It is natural to assume that their samples on SiO₂ have an inhomogeneous surface structure. The authors can categorize the dI/dV spectra based on the shape of band tails and make a correlation with the STM image.

8) The authors only discussed the structural disorder effects on the LDOS of a 2D semiconductor MoS₂, but the SiO₂ substrate includes many impurities leading to the inhomogeneous electronic structure in the 2D material. How do they exclude this possibility?

9) In Fig. 5, the authors showed that the curved atomic structure leads to a peculiar magnetic structure. Do their experimental results exist to support this theoretical expectation?

Reviewer #1

In this work, Bong Gyu Shin and co-workers studied the quantum fluctuations in disordered 2D semiconductors. By means of gate-tunable STM and STS measurements, the authors demonstrate that the curvature of MoS₂ localizes the charge to weaken the strain-induced band gap renormalization, leading to band edge flattening, which is further supported by theoretical calculations. The study of critical energy with the crossover between localized and extended states also offers great potential in understanding the metal-insulator transition in disordered 2D semiconductors. In addition, the authors explore the disorder effect in local magnetic moment due to the change of local spin orbit coupling. I recommend the publication of this work in Nature Communications. Before that, addressing the following issues can further strengthen the paper.

1. The authors need to specify the method/details on how to determine the energy position of the band edges with band tails?

Author Reply: Following the reviewer's comment, we have added the details on how to determine the band edges in "Supplementary Fig. 6" as follows:

In the caption of Supplementary Fig. 6,

b, Determination of band gap in STS spectra. The band edges (or band gap) were determined by linear fitting of $\log(dI/dV)$ (dashed lines) in the energy range of $E_{2\sigma, c(v)}$ to $E_{2\sigma, c(v)} + (-) dE$, where $E_{2\sigma, c(v)}$ is the energy point of intersection between $\log(dI/dV)$ and a level that the average of the background noise (C_{bg}) plus twice the standard deviation of the background noise (2σ) in the band gap region (green line), and dE is 300 meV. The band edges were recorded at the intersections of the linear fits and C_{bg} (purple line). These are indicated by red dots on the plot. The blue and red shaded regions indicate the energy ranges (dE) for the linear fitting. This empirical determination procedure of band gap was suggested in the ref. 68.

In addition, the experimentally observed STS in randomly deformed monolayer MoS shall be presented in parallel with the theoretical DOS using tight-binding methods (Figure 4i). In addition,

Author Reply: The experimental results for the band tails with the characterization of surface morphologies have been added in "Supplementary Fig. 4" (as below)

Supplementary Fig. 4. | Band tail formation in structural disorder. Rough MoS₂ surface with higher curvatures shows larger band tail width in comparison with the band tail in the flat region. $\langle dI/dV \rangle$ indicates the spatial average of dI/dV results corresponding to the density of states. The flat region (denoted as 'flat') shows a larger band gap with negligible band tail widths, which is close to the intrinsic property of monolayer MoS₂. The band tails of the regions labeled '1' and '2' in the insets exhibit significantly different widths. The average values of mean curvatures in the '1' and '2' regions are 0.1593 and 0.1867 nm⁻¹, respectively. Each curvature value corresponds to the bending strain of 2.632 and 3.111%. The higher value of curvature induces a more protruded band tail and larger band tail width. The arrows indicate protrusions of band tails.

All results align well with the main conclusions in the manuscript.

2. Some localized states might be linked to the surface morphology or curvature. Have the author considered other possible sources of localized states near band edges, such as sulfur vacancies or charge impurities on SiO₂.

Author Reply: Defects are distinguishable in STM images and STS spectra. Sulfur vacancies have an in-gap state and specific topological features with a size of less than 2 nm. Most other defects also show in-gap states and clear topological features with dimensions of around 2 nm. However, the curvature-induced band gap fluctuation and charge localization are smoothly distributed over at least ~4 nm, which is a significantly larger characteristic length scale than those of defects. Therefore, we ruled out defects as the origin of the observed band gap fluctuations and charge localization.

Charge impurities also show distinguishable STS spectra and topological features. The charged impurities act like extrinsic potentials and the band bending direction is dependent on the charge states of the impurities. The STS spectra affected by positively charged impurities show the rapid upward band bending that exhibits a large change in the local Fermi level position compared to the conduction band edge. Such change in the local density of states is rapid and sharp, which is different from the smooth change of the flattening of band edges via doping. Moreover, the features of charge impurities are small and sharp. Therefore, we can clearly distinguish the effects of charge impurities and curvature.

Below are the STM/STS results and DFT calculations of defects and charged impurities,

Characterization of defects and charged impurities. **a-c**, Atomic resolution STM images with defects in MoS₂ on SiO₂. The defects show the bright and dark spot features in a 1-2 nm length scale. (sample bias: -3 V (**a**) and -2 V (**b,c**)). **d-e**, dI/dV line profiles for a defect (**d**) show an in-gap state and band gap reduction along with the surface morphology. The defect exhibited the small feature at the top of the height profile (**e**) with a radius of curvature of 9.2 nm, indicated by a red-dashed line, induced by the band gap reduction. **f**, A single STS spectrum of the defect in (**e**). The arrow indicates the in-gap state of the defect. **g-h**, Observation of charged impurities. STM (**g**) and corresponding STS mapping (**h**) results for electron-doped MoS₂ on SiO₂ with charged impurities (sample bias: -3.5 V). The STM image (**g**) shows additional adsorbed particles (indicated by arrows). The corresponding STS mapping results (**h**) at the sample bias of 1 V show dark regions exhibiting rigid shifts of the STS results on the axis. If bias. **i**, STS results at the numbered spots in (**h**). The '0' spot did not involve any effects of charged impurities. The '1'-'4' spots exhibited the shifting of dI/dV spectra by the extrinsic potentials of charged impurities. The '1' and '2' spots included the adsorbed particles on MoS₂, but the '3' and '4' spots were affected by charged impurities between MoS₂ and the SiO₂ surface. **j,k**, DFT calculations for the charged impurities. The positive (negative) polarity of the impurities was modeled by a proximal effect of O (K) atom. By the potential of the positively (negatively) charged impurity, band bending shows an upward (downward) rigid shift near the charged impurity as shown in (**j**) ((**k**)). Such dramatic band bending by extrinsic potentials is different from the flattening of band edges via doping. The change of band edges by the extrinsic potentials occurred in one direction, which is different from the curvature-induced band gap fluctuation that involves opposite directions in changes of band edges (*i.e.*, reduction of CBM and increase of VBM). By the distinguishable features of defects and charged impurities, we ruled out the defects and charged impurities from the analysis of curvature-induced band gap fluctuations and charge localization. (An additional feature in the band gap regions of MoS₂ ((**j**)) and (**k**)) is an artifact by the modeling, originated by O or K atom, which does not invalidate the central conclusion.)

Can the authors estimate the density of defects and comments their impact on the electronic disorder?

Author Reply: We statistically obtained the defect density of $2.91 \pm 0.11 \times 10^{11} \text{ cm}^{-2}$ from STM results. The impact ranges of defects were 1-2 nm scale as observed in the above figures. A localization/correlation length induced by defects near a critical defect density is, therefore, expected to be shorter than that of the curvature effect. In the observed defect density, the distance between defects has far larger than their impact ranges. The existence of a critical defect density and the effect of defect-induced scattering with a short correlation/localization length should be investigated further.

3. The occurrence of MIT has not been confirmed experimentally because the gate injected carrier density does not reach the critical doping. In this context, some sentences (line 22-25) in the abstract and line 66-71 might be misleading.

Author Reply: For clarification, we have specified the observation of quantum fluctuations in wavefunctions as “the localization-delocalization transition” instead of ‘MIT’ in line 1 on page 2 and in line 18 on page 3.

4. In line 149, authors conclude that “the variation of band edges is saturated at a low value, indicating the spatial flattening of the band edge”. However, it seems that the saturation is due to the heavily doping that the Fermi level has already touched the band edge rather than the band edge flattening. The authors should compare the band edge energy with MoS2 in the same doping level but without curvature.

Author Reply: The flattening of band edges in Fig. 2h started to saturate at $\sim 5 \times 10^{13} \text{ e/cm}^2$. However, the relative energy position of the conduction band edge compared to the Fermi level ($CBM - E_F$) is still decreasing for both flat and deformed (cylindrical) cases, which is not saturated (see the graph below), even at a high doping level of $\sim 10^{14} \text{ e/cm}^2$.

The energy difference of $CBM - E_F$ between flat and deformed cases was caused by the preference for filling on the curvature regions. After the Fermi level touches the conduction band minimum, the volume for the filling was reduced in the deformed case that the states near band edges on curvature regions had been already occupied, enhancing the change of the Fermi level via doping up to $\sim 3 \times 10^{14} \text{ e/cm}^2$. Eventually, the energy difference of $CBM - E_F$ between flat and deformed cases for the higher doping becomes smaller, involving the flattening of band edges in the deformed case via doping.

The origin of the flattening band edges via doping is the charge screening effects of potentials on curvature regions (lower potential sites) by charge accumulation, which differs from the heavy doping saturation.

We have added an additional figure in “Supplementary Fig. 3”, presenting the relative energy position of the conduction band minimum to the Fermi level as a function of doping concentration.

Supplementary Fig. 3. | DFT calculations of the relative energy position of the conduction band minimum (CBM) to the Fermi level as a function of doping.

The Fermi level (E_F) changes continuously up to the high doping level of $\sim 10^{15} \text{ e/cm}^2$. The difference in $CBM - E_F$ between the cylindrical curvature structure and the flat structure became significant after the E_F reached the CBM (i.e., $CBM - E_F = 0$) at the doping level above $\sim 2 \times 10^{13} \text{ e/cm}^2$. This is attributed to the band tails near the band edges, which are already occupied to reduce the effective capacity of doping charge above the conduction band edge. Therefore, the Fermi level change is enhanced in the curvature structure. Above the $\sim 3 \times 10^{14} \text{ e/cm}^2$, however, the $CBM - E_F$ of flat and cylindrical structures become similar again including the effect of the flattening

of band edges which started to be saturated at $\sim 5 \times 10^{13} \text{ e/cm}^2$. The inset shows the neutral states ($\Delta n_e = 0$) of both cases. The smaller value of $CBM - E_F$ for the cylindrical case at the neutral state is due to the curvature-induced band gap fluctuation.

5. It is not very clear how the authors figure out the relationship between surface morphology and curvature as shown in Figure 3c, d. Should the hill and valley regions have the opposite sign of curvature? The authors need to explain this.

Author Reply: The ‘absolute’ values of (mean) curvatures are the main causes of band gap fluctuations. The curvature of hill and valley regions in Fig. 3d are all positive since Fig. 3d shows average values of the absolute curvatures. Each average value (Fig. 3c-e) was obtained within the range of 12 nm as shown in Fig. 3a and b. We have corrected the axis label in Fig. 3d to ‘absolute curvature’. In addition, we have added the following sentence in the caption of Fig. 3; “The average values for plots were obtained in the axis of 12 nm in (a) and (b).”.

Taking the reviewer’s comments into account, we have presented the overall plot for each line profile of curvatures in “Supplementary Fig. 2”. In this plot, each line profile of curvature has opposite signs for the hill (–) and valley (+) regions.

Supplementary Fig. 2. Total curvature plot from Fig. 3a. Each line profile of curvature in Fig. 3a was plotted. The average value of the absolute curvatures was given in Fig. 3d.

6. How to explain the fine features in Fig.3h? And why the electron and hole doping has an unsymmetric effect to the local work function variation (hole doping has fine features but electron doping does not) ?

Author Reply: The fine features in curvature regions of the electron-doping cases are caused by the preference of electron filling on each atom. The electron-doping charge firstly prefers to occupy Mo atoms near the conduction band edge, which lowers the local work function more than that of S atoms. (The higher electron density lowers the local work function.) After the full occupation of Mo atoms near the local conduction band edge in the preference of curvature effects, electron filling in S atoms eventually become dominant. That is why slightly higher fine features of local work functions in curvature regions are shifted from S to Mo atom positions with increasing electron-doping concentration. Additionally, the fine features of curvature regions are reduced and their background is flattened when electron-doping is increased due to the charge screening effects, as shown in Fig. 3e and h. This means that the binding energy difference of electrons between Mo and S atom sites becomes smaller, because of the charge screening from the accumulated charge. The charge screening also flattens the local work function in the high curvature area where charge density is high.

In the hole-doping case, however, fine features and flattening do not exist in the curvature regions, since both Mo and S atoms have a larger capacity for hole doping near the valence band edge. The reversed trend of local work function fluctuation between the electron and hole doping cases is attributed to the opposite signs of carrier charge. In other words, the positive and negative charges attract each other, and the binding energy for electrons becomes higher corresponding to the higher local work function.

Considering the reviewer’s question, we have added a further explanation for local work function fluctuation in line 11 on page 9 of the manuscript:

“..., reversing the overall trend of fluctuation in local work function compared to the electron-doping case. In addition, the fine features of local work function in the curvature regions for the electron-doping case are shifting from S to Mo atom sites with increasing the electron-doping level along with the flattening of both background and fine features in the curvature regions as shown in Fig. 3h. The hole-doping cases did not show fine features and flattening of local work function in the curvature regions. The difference originated from orbital configurations near the band edges which differentiate the density of states and the capacity of charge occupation (see also Supplementary Fig. 1 and 3).

For clarity, we have detailed the symbol of the charge in the caption of Fig 3.

“, where e is the electron charge and the sign of + (–) is the addition (subtraction) of electrons.”

Additionally, we have added an additional figure in “Supplementary Fig. 1”.

Supplementary Fig. 1 includes doping charge densities and integrated DOSs near each band edge, showing a larger capacity of states near the valence band edge.

Supplementary Fig. 1 | Asymmetry of the density of states in deformed MoS₂. **a**, Doping charge density near the conduction band edge with the energy range of 0.3 eV. **b**, Doping charge density near the valence band edge with the energy range of 0.3 eV. The electron-doping concentration in (a) and (b) is $13.8 \times 10^{13} \text{ e/cm}^2$. The isosurface of charge density in (a) and (b) is 10^{-4} e/\AA^3 . **c**, Integrated DOS near the band edges for the intrinsic case. The integration of DOS was performed from the Fermi level to each band side. The yellow shade shows the difference between conduction and valence bands near the band edges. The energy is an absolute value. The zero energy is the Fermi level.

7. In Fig. 4d and line 214, authors conclude that "the uniform intensity regardless of the structural disorder is a result of the band edge flattening", but in Fig. 1f it shows that the extended states still show a big variation.

Author Reply: The band edge flattening was induced by doping charge. The doping charge can screen the structural-disorder-induced potential fluctuation, resulting in band edge flattening.

Fig. 1f shows the intrinsic (neutral) states of MoS₂ (as an STS map) above the conduction band edge in the disordered MoS₂. In the intrinsic case without doping, the electronic structures exhibited structural-disorder-driven fluctuations over a wide range of energy, not only for the band edges. Theoretical results (Extended data Fig. 4a-d) also reproduced the random fluctuations of states caused by structural disorders, which is in good agreement with experimental results.

Fig. 4d shows the STS map of the high electron-doping case above the conduction band edge, which clearly proves the conduction band edge flattening.

To provide further clarity on this point, we have modified the manuscript,

In the caption of Fig. 1, we have added "for the intrinsic state of disordered MoS₂", subtitles "intrinsic (neutral)" for Fig. 1d-g, "electron-doping" for Fig. 1j, and "hole-doping" for Fig. 1m.

In line 17 on page 11, we have added "by the doping charge".

8. A discussion of the experimental realization for the curvature-induced magnetism in disordered 2D semiconductors shall be included.

Author Reply: The intrinsic magnetism of monolayer MoS₂ on SiO₂ was already reported in the previous report (ref. 38), however, the origin of the intrinsic magnetization was not yet understood. Our DFT results of structural models align well with a previously reported experimental value of $\sim 4.3 \times 10^{-7} \mu_B/\text{\AA}^3$ and prove that the origin of the intrinsic magnetism is the structural disorder. Moreover, the result also agrees well with the average of Gaussian curvature ($\kappa(\mathbf{r})$)-induced pseudo-magnetic fields from the STM result ($\sim 4.0 \times 10^{-7} \mu_B/\text{\AA}^3$), which supports the curvature-induced intrinsic magnetism further. (The average of curvature-pseudo-magnetic field relation: $\langle M \rangle = (\chi_{\text{MoS}_2}/\mu_0) \langle \mathcal{B} \rangle = (\chi_{\text{MoS}_2}/\mu_0) \langle \frac{\hbar}{2|e|} \kappa(\mathbf{r}) \rangle$, see Supplementary Discussion.). In addition, the previous study reported that the curvature-induced pseudo-magnetic field reproduced the experimental observation of the peculiar negative magnetoconductance in monolayer MoS₂.

On the other hand, if one considers the defect states as a source of magnetism, the observed defects in STM/STS results did not correspond to the previous results of magnetization. Most defects' configurations were predicted to be non-magnetic. Mo substitution with S and Mo vacancy are possible sources for the magnetization but they were not observed in the STM/STS results.

Taking the reviewer's comment into account, we have added further discussion in the manuscript,

Line 28 on page 15:

“Furthermore, the calculated magnetoconductance in the curvature-induced pseudo-magnetic field agrees well with the previous experimental results of the negative magnetoconductance in monolayer MoS₂ (ref. 45).”

Some minor issues to be addressed:

9. *In Fig. 4j, are three data points sufficient to demonstrate the correlation between the standard deviation of the band gap distribution and band tail width?*

Author Reply: Experimental data points in Fig. 4j reflect a statistical analysis of a very large amount of data: ~10⁴ STS spectra were obtained for one data set in Fig.4j. All three experimental data points are surprisingly well aligned with theoretically extracted values. In addition to the ~4×10⁴ theoretical LDOS spectra for each data point, three experimental data points are sufficient to confirm the agreement between theory and experiment.

In the caption of Fig. 4., we have added sentences below,

“Each data point in (j) was obtained out of the several data sets. And each of the data sets includes over the ~10⁴ spectra.”

10. *Authors should mark the gate voltage for electron and hole doping in Fig. 1j and 1m, and estimate the charge carrier density based on the sample configuration.*

Author Reply: Following the reviewer’s comment, we have added gate voltages and estimated charge carrier density in the caption of Fig. 1,

“... (j) Band edges at electron-doped state ($\Delta n_e = 5.67 \times 10^{12} \text{ cm}^{-2}$ (gate bias of 75 V)). Schematic plots of (k) doping charge localization and (l) its local band bending. m-o, Spatial flattening of VBM via hole-doping. (m) Band edges at hole-doped state ($\Delta n_h = 4.54 \times 10^{12} \text{ cm}^{-2}$ (gate bias of -60 V)). ($\Delta n_{e \text{ or } h}$ denotes electron (or hole) doping concentration.) ...”.

11. *The authors should specify the sample bias they applied for measuring the TBH.*

Author Reply: We have specified the sample bias information;

In the caption of Fig.3: “The sample bias of -3 V was applied.”

In the caption of Extended Data Fig. 5.: “The sample bias of -2.5 V was applied.”

Reviewer #2

Compared with traditional semiconductors, 2D semiconductors are flexible and atomically thin and can adjust its surface profile to the landscape of substrates. This makes the metal-insulating transition (MIT) mechanisms in 2D semiconductors distinct from traditional semiconductors. Although many works have been conducted to study this effect in 2D materials, majority of them are from a macroscopic point of view, such as transport and capacitance, and the driving mechanisms are debating and not well understood. In this work entitled “Structural-disorder-driven critical quantum fluctuation and localization in two-dimensional semiconductors”, Shin et al. investigated the metal-insulating transition mechanisms in 2D semiconductors from the microscopic point of view. Direct STM and STS are performed in a wide members of TMDC family on substrates, including MoS₂, WS₂, etc.. Random curvature in TMDS/SiO₂ induced band reconfiguration and localization states are observed near the band edge. Electron/hole Doping will flat the band edge of conductance/valence band and extended states will dominate. The observations can well explain the MIT in 2D materials and agree with the overall picture of percolation-transition. DFT calculations also well support the experimental observations. Overall, the topic about MIT is interesting and attracts researchers’ attention. The experimental results, theory calculation and analysis are solid. I recommend for publication after addressing my concerns.

1.As shown in the introduction part “MoS₂ encapsulated in hBN does not show MIT, since hBN is very flat”. Based on your proposed theory, what’s the critical “defect density” (more precisely, the critical degree of curvature variations in MoS₂) that allows MIT occur in MoS₂?

Author Reply: If one considers the random fluctuations of surface morphology as a collection of curved features with Gaussian shape, we can analyze the fluctuation with ‘density of features’ and ‘distribution of curvature’, as long as the features are distributed and do not overlap. A lower density of bumps reduces the probability of transport or tunneling to other bump regions in a way of percolation. A higher curvature of bumps provides a deeper potential well, which makes it harder for the carriers to escape to other regions. Therefore, both ‘density of bumps’ and ‘distribution of curvature’ will affect transport. As far as we know, the critical degree of curvature variation that allows MIT is not reported yet. We believe finding this critical degree is a very important future project.

2.Similar to the first question, based on my understanding of your proposed model, as long as there is random curvature in MoS₂, there will be localization states near band edge. Flatter surface will result in narrower band tail and lead to smaller E_c . If it is true, MIT should be universal in MoS₂ but with variation of critical MIT transition energy/carrier density, since there is no perfect MoS₂ flakes. Is this correct?

Author Reply: Yes, the ‘universality’ is the main reason why we think curvature-induced MIT is so important. However, the critical transition energy or critical carrier density will be different for a sample and dependent on various extrinsic factors inducing structural disorder. The curvature-induced localized states form band tails near the band edges and they can be assigned as a charge trap or trap capacitance for transport.

3.Please discuss on the MIT transition in thicker samples based on your proposed theory and compare with previous experimental results (Nat Commun 6, 6088, 2015) if possible.

Author Reply: The previous experimental results in *Nat. Commun.* 6, 6088 (2015) provided evidence for charge traps near the band edges and discussed the trend of the critical doping density depending on the number of layers or effective thickness. In the literature, as increasing the number of layers, the critical doping density tends to be smaller. The observed charge trap density for monolayer MoS₂ was larger than that of trilayer MoS₂. Those results are compatible with our main conclusions. The formation of curvature depends on the flexibility of the material. A thinner layered material is more flexible for bending. For example, monolayer MoS₂ more easily conformed to the surface roughness of a substrate than that bilayer MoS₂ did. The observed surface roughness of monolayer MoS₂ on SiO₂ was larger than that of bilayer MoS₂ on SiO₂. The surface roughness of trilayer MoS₂ is smaller than that of monolayer MoS₂, leading to a smaller curvature formation. The smaller curvature provides a narrower fluctuation range of band edges inducing less capacity for charge localization near band edges. Then, the smaller capacity of charge localization was fulfilled with a smaller critical doping density, which is in good agreement with the previous results in the literature. (the reported values for critical doping density; for monolayer: $\sim 1 \times 10^{13}$ cm⁻², for trilayer: $\sim 8.6 \times 10^{12}$ cm⁻², and for multilayer (~ 20 nm): $\sim 6.7 \times 10^{12}$ cm⁻²) In addition, the localized charge by the curvature mechanism can explain why the percolation-type metal-insulator transition by thermal activation occurs.

In line 26 on page 13 of the manuscript, we have added the discussion below;

“In addition, the curvature-induced charge localization explains why the charge trap near band edges and critical doping charge density became smaller as increasing the thickness of MoS₂ (multilayers) in the previous results³⁹. As increasing the thickness, the flexibility of MoS₂ decreases leading to smaller curvature formation allowing less capacity for charge localization (Extended Data Fig. 3).”

4. Other defects, such as vacancies, also exist in MoS₂. How to understand the “joint” effect of these defects and your proposed curvature-induced localization to MIT.

Author Reply: Most of the defects show in-gap or additional states and cause local band bending in a different way compared to the curvature effects. The observed defects were distinguishable in STM and STS results. Defects show typical dimensions of ≤ 2 nm and in-gap states with different band bending behaviors as shown below;

The defect features (a-e) in STM/STS results are smaller than the typical scale of the curvature effects. As shown in the above figures d and e for a defect, band gap reduction was affected by the curvature of the hill in MoS₂ (the radius of curvature is 9.2 nm as high curvature, red dashed line in (e)). The defect induced a small morphological protrusion at the hill (e) and exhibited an additional in-gap state along with band bending as a rigid downshift near the defect as shown in (d). The rigid downshift corresponds to a positive charge at the defect bringing the local Fermi level closer to the conduction band edge. The charged states of the defect determined the band bending behavior. Such band bending by the defect is different from the curvature-induced band edge changes that shows the opposite directions in changes of each band edge (i.e., lowering of CBM and up-lifting of VBM). The results show overlapping of the effects from curvature (long range) and defect (short range). Figure f shows a single STS spectrum on the defect and an arrow indicates the in-gap state.

From the observations in STM/STS results, the ‘joint’ effect of defects and curvature is expected to be complex, resulting from both short- and long-range interactions. As shown in the above figures a-d, the impact range of a defect is 1-2 nm. A correlation/localization length induced by the defects near a critical defect density is, therefore, expected to be short. The existence of a critical defect density and the effect of defect-induced scattering with a short correlation/localization length should be investigated further.

5. Previous STS/STM works on graphene have shown puddle-like structures. Could you please compare the difference between your theory and theirs.

Author Reply: The puddle-like features in graphene were explained by charged impurities. The charged impurities act like an extrinsic potential inducing a rigid shift in one direction as a local band bending. The band bending direction is dependent on the charge states of impurities, which is down(up)-shift for a positive (negative) charge. The random fluctuation of charged impurities shows charge-dependent random band bending behaviors. The curvature-induced fluctuations of band edges or band gaps involve opposite directions in changes of band edges (i.e., reduction of CBM and increase in VBM). Such fluctuations differ from the band bending of one-directional shifts by extrinsic potentials or charged impurities.

In our observations, charge impurities show unique features in STS spectra. The feature sizes of charge impurities are small (~a few nanometers) and sharp. The STS spectra affected by negatively charged impurities show the rapid upward band bending that exhibits a large change in the local Fermi level position compared to the conduction band edge. Such change in the local density of states is rapid and sharp, which differs from the smooth change of the flattening of band edges via doping as shown below;

g-h, Observation of charged impurities. STM (**g**) and corresponding STS mapping (**h**) results for electron-doped MoS₂ on SiO₂ with charged impurities. The STM image (**g**) shows additional adsorbed particles (indicated by arrows). (sample bias: -3.5 V) The corresponding STS mapping results (**h**) at the sample bias of 1 V show dark regions exhibiting rigid shifts of the STS results on the axis of bias. **i**, STS results at the numbered spots in (**h**). The ‘0’ spot did not involve any effects of charged impurities. The ‘1’-‘4’ spots exhibited the shifting of dI/dV spectra by the extrinsic potentials of charged impurities. Spots ‘1’ and ‘2’ included the adsorbed particles on MoS₂, but spots ‘3’ and ‘4’ were affected by charged impurities between MoS₂ and SiO₂ surface. **j,k**, DFT calculations for the charged impurities. The negative (positive) charge type of impurities was modeled by a proximal effect of O (K) atom. By the potential of the negatively (positively) charged impurity, band bending shows an upward (downward) rigid shift near the charged impurity as shown in (**j**) ((**k**)). Such band bending by extrinsic potentials is different from the flattening of band edges via doping. The change of band edges by the extrinsic potentials occurred in the same direction, which is different from the curvature-induced band gap fluctuation that involves opposite directions in changes of band edges (*i.e.*, lowering of CBM and up-lifting of VBM). By focusing on the distinguishable features of defects and charged impurities, we were able to exclude these defects and charged impurities in the analysis of curvature-induced band gap fluctuations and charge localization.

In addition, the puddle-like features in graphene were also manifest in substrate-induced deformation. The deformation in graphene leads to the variation of overlapping between the π orbitals (p_z) of the carbon atoms. In the continuum limit, such strain induces an effective gauge field with scalar and vector potentials. A strained MoS₂ is known to be a different gauge class from graphene. The effective gauge fields in deformed MoS₂ are under investigation.

In line 15 on page 5 of the manuscript, we have inserted the sentence that;

“In the STM/STS results, the curvature-induced effects were clearly distinguished from the defects and charged impurities.”.

6. Please show more physical meaning of R and “autocorrection” shown in Fig. 4, which is not clear enough to me.

Author Reply: In general, the autocorrelation of a physical quantity shows self-similarity or self-affinity, which can be used to analyze uniformity or pattern in the distribution of the physical quantity. The autocorrelation consists of the product of values of two points having distance R . The autocorrelation of STS mapping results is related to the localization or correlation length of wavefunctions. If an extended state is given, the wavefunction of the state shows a large correlation length due to the spatial coverage of the wavefunction that reaches the (effective) system size. Such wide coverage of wavefunctions leads to conductivity, indicating that the probability for a finding of charge carriers is large enough from one boundary to the other boundary of the system as a transport path. Therefore, the wavefunctions for conductance are sensitive to the boundary conditions, for example, electrode connections with a bias.

If localized states were given, the wavefunctions of the states show a shorter localization length than correlation lengths of extended states due to the decaying (peak) shape of localized states that limits the self-similarity range of each localized state at a position, indicating that self-resemblance could be only found near the center of localized states by an overlap between the wavefunctions and translated wavefunctions of themselves. Localized states are isolated at a position, not sensitive to the boundary conditions, leading to insulation.

The long-tailed high value in the radial-averaged autocorrelation from LDOS maps indicates a uniform extended state, whereas a rapid exponential decay and low value in the radial-averaged autocorrelation indicate a localized state. In between localized and extended states, a power law behavior in the radial-averaged autocorrelation is

involved near/at the criticality. The critical wavefunctions are fluctuating but extended, exhibiting an irregular pattern with a multifractality under disorder. For the characterization of the critical energy, the important key is a power law behavior in the radial-averaged autocorrelation of an LDOS map.

For further discussion, multifractality is important. The fractality involves the “absence of length scales” associated with a power law; for example, when someone magnifies a portion of a fractal image, the same pattern before magnification can be observed again. For a physical quantity $g(L)$ as a real-valued function of a scale L , if scaling by a factor s follows $g(sL) = s^k g(L)$ where k is an exponent, then a solution of $g(L)$ has the form of $g(L) \sim L^k$, a power law. If the g and k are a functional of powers q , multifractality means significant non-linearity of $k(q)$. With the multifractality, the scaling behavior of g is anomalous.

The shapes of the critical wavefunctions should be extended without a characteristic length (absence of length scales), otherwise, the critical wavefunctions for the localization-delocalization transition depend on the system size compared to the localization/correlation length of the critical wavefunctions, i.e., a larger correlation length than system size leads to the conductivity of the system, or a shorter case is insulating. Such scale dependence for the criticality cannot be treated as a general condition or definition for localization-delocalization transition. The critical wavefunctions should be capable to cover any system size for the generic definition. The absence of length scales in the critical wavefunctions helps the criticality stay in the general definition for the critical states.

The signature of the multifractality at the critical energy is associated with the radial-averaged autocorrelation of LDOS, which have been investigated theoretically. The characterization of multifractality can be performed by the moment method including the mass exponents ($\tau(q)$), coarse Holder exponent ($\alpha(q)$), and the singularity spectrum ($f(\alpha)$) as described in Methods in the manuscript. The radial-averaged autocorrelation of LDOS near the critical point shows a scaling law (a power law),

$$\langle \rho(E, \mathbf{r}) \rho(E, \mathbf{r} + \mathbf{R}) \rangle / \langle \rho(E) \rangle^2 \propto (\mathcal{L}/|\mathbf{R}|)^\eta, \quad l < |\mathbf{R}| < \mathcal{L},$$

where the exponent $\eta = d - \tau_2 \equiv -\Delta_2$, and $\langle \rho(E) \rangle$ is the disorder-averaged LDOS at the energy E that is counted from the electrochemical potential (Fermi level), l is the length scale of elastic scattering mean free path, $\mathcal{L} = \min\{\xi, L_\phi, L\}$ is the shortest length as an effective system size among the localization or correlation length ξ , dephasing length L_ϕ , and system size L . The relation $\eta = -\Delta_2$ engages multifractality and autocorrelation near/at criticality.

In line 23 on page 11, we have added a sentence below,

“In the radial-averaged autocorrelation results, the rapid decay into a lower value indicates localized states and the slow decay with a higher value corresponds to the extended states.”

7. Fig. 1i, k, n are theoretical calculated or schematic. Please put clearly in the legend.

Author Reply: We have added ‘schematic plot’ in Fig. 1i, k, and n.

8. Since the manuscript is relatively long, I suggest to put some subtitles to increase the readability of the manuscript.

Author Reply: We have added the subtitles in the manuscript;

“**Structural-disorder-driven charge localization**” in line 1 on page 5,

“**The flattening of band edges via doping**” in line 16 on page 5,

“**Variation of local work function with charge localization**” in line 13 on page 8,

“**Criticality in quantum fluctuation of wavefunctions**” in line 2 on page 11,

“**Band tail formation in structural disorder**” in line 27 on page 12,

“**Structural-disorder-driven magnetism**” in line 10 on page 14,

“**Summary and conclusions**” in line 14 on page 16.

Reviewer #3

The authors studied the inhomogeneous electronic structure in a corrugated 2D semiconductor MoS₂ using gate-tunable scanning tunneling microscopy and spectroscopy. They have observed that the structure disorder results in the band gap's spatial variation, which can be flattened by electron(or hole)-doping, and the localized states forming band tails. In addition, the autocorrelation functions and histograms of the energy-dependent LDOS maps exhibit the power-law behavior and the log-normal distribution that have been theoretically expected as a hallmark for the quantum criticality of metal-insulator transitions. Although they do not provide a new perspective for the main point, their results will be constructive in understanding the electronic properties of flexible 2D semiconductors. However, given the informative results in this field, the following issues should be carefully addressed before I can recommend its publication in Nature Communications.

1) They used STM images to determine the curvature of the curved atomic structure. However, the contrast (z-height) of the STM image is determined by the atomic and electronic structure of the sample. A spatially inhomogeneous electronic structure also leads to contrast variation in the STM image. Please address how authors can exclude or minimize this effect.

Author Reply: STM measures isosurface of electron charge density in convolution of orbitals between tip and surface. There are many factors affecting the fluctuation of the charge density isosurface. Typically, each atom in a short distance or atomic-scale defects (for example, vacancy, charged impurities) induced fluctuations of the isosurface on a length scale of tens of picometers. For the analysis of curvature effects, we focused on the positional changes of each atom (curvature or z-height change) on the nanometer scale. The curvature-induced isosurface fluctuations differs by about 100 times (~1 nm/(10 pm)) in comparison to the atom-induced protrusions. Therefore, we can exclude the orbital scale corrugations of each atom in a short distance range (interatomic scale (Å)) and focus on the curvature effect on the positional change of atoms over a large distance range (> 1 nm). Inhomogeneous electronic structures can be triggered by defects and charged impurities. However, the impact distance range of them and signatures of electronic structures are narrow and distinct. Defects and charged impurities show distinguishable features in STM images and STS spectra (please see more details in Author reply to issue 8 later), which can be clearly identified and excluded when we analyze the curvature effect.

In “Supplementary Fig. 6a”, we have added an example showing a typical curvature fitting of STM line profile.

Supplementary Fig. 6a. | Characterization of curvature and band gap. a, A typical line profile of deformed MoS₂ with a curvature. The dashed red line is a fitting result of the hill feature in the height change of MoS₂. The STM results show isosurface of electron charge density which is related to the positional changes (of nuclei) and orbitals of each atom. Small corrugation of picometer scale corresponds to the atomic protrusions of S atoms, which were not considered for the curvature analysis.

2) They did not show full-range dI/dV spectra. It is necessary to offer a few representative dI/dV curves and explain how to determine the band gaps to construct the images shown in Fig. 1c.

Author Reply: Following the reviewer's comment, we have described details for how the band edges is determined in “Supplementary Fig. 6b” as below;

b, Determination of band gap in STS spectra. The band edges (or band gap) were determined by linear fitting of $\log(dI/dV)$ (dashed lines) in the energy range of $E_{2\sigma,c(v)}$ to $E_{2\sigma,c(v)} + (-) dE$, where $E_{2\sigma,c(v)}$ is the energy point of intersection between $\log(dI/dV)$ and a level that the average of the background noise (C_{bg}) plus twice the standard deviation of the background noise (2σ) in the band gap region (green line), and dE is 300 meV. The band edges were recorded at the intersections of the linear fits and C_{bg} (purple line). These are indicated by red dots on the plot. The blue and red shaded regions indicate the energy ranges (dE) for the linear fitting. This empirical determination procedure of band gap was suggested in the ref. 68.

3) Their DFT calculations showed the band gap variation determined by the curvature of the atomic structure for several semiconducting TMdC materials in Fig.2. Here, I cannot find the specific reason why it is necessary to include other materials. It would be better to focus on MoS₂ in the main text and move the results of other compounds to the supplementary.

Author Reply: The DFT calculations in Fig. 2 demonstrate the **universality** of curvature-induced band gap fluctuations and band edge flattening in 2D semiconductors. Below are the general properties of 2D materials, which are the important cause of curvature-induced band gap fluctuations.

1. Practically, perfectly flat 2D semiconductors are almost unrealistic, as shown in Extended Data Fig. 3.
2. Compared to the 3D materials, 2D materials are more flexible and their electronic structures are more sensitive to the bending strain. Therefore, the extrinsic factors, such as substrates and adsorbates, affect the 2D materials significantly.
3. Geometrically, 2D semiconducting materials are less effective in dielectric screening of Coulomb interactions. Therefore, 2D materials have strong Coulomb interaction, resulting in the enhanced band edge flattening.

Those are contrasts to the 3D bulk semiconductors, which are difficult to deform locally and have weak Coulomb interactions inside the materials. Those differences can explain the strong binding energy of excitons in TMDs compared to the bulk Si. Therefore, structural disorder is not considered as a major reason for MIT in 3D bulk materials. However, the understanding of the curvature effect in 2D semiconductors is essential.

We proved that many other 2D semiconductors show similar phenomena of band gap fluctuations based on the same mechanism. Only the degrees of variations can be differed depending on the material properties.

In line 25 on page 7, we have emphasized those points as below,

“Generally, 2D materials are flexible and their electronic structures are sensitive to the extrinsic factors such as a substrate and adsorbates. Moreover, 2D materials have strong Coulomb interactions due to poor dielectric screening by a small volume of themselves, which enhances the band edge flattening. Therefore, the curvature-induced band gap fluctuation and band edge flattening by doping are universal in 2D semiconductors. The degree of variations in band gap and band edge flattening depends on material properties.”

4) Their DFT calculations show that CBM exhibits more pronounced spatial variations than VBM in the cylindrical curvature structure. The physical reason for the asymmetry needs to be better explained in the main text.

Author Reply: The flattening of band edges via doping is a local band bending achieving equilibrium of electrochemical potential. The equilibrium of electrochemical potential is related to change in total energy of electrons when the addition or removal of electrons. One of the reasons for asymmetry in flattening of band edges is different orbital configurations near each band edge that differentiate the local band bending in the equilibrium of electrochemical potential. The conduction band edge mainly consists of *d* orbitals of Mo atoms and at higher energy level above the conduction band edge, *p* orbitals of S atoms is involved together with orbitals of Mo atoms. The valence band edge involves the *d* and *p* orbitals together. As a result, the capacity for occupation of the states near each band edge is different. The local density of states near valence band edge is larger than that of conduction band edge, which is the one of the reasons for asymmetry of the flattening of band edges.

In “Supplementary Fig. 1”, we have added theoretical results to explain the asymmetry of DOS in deformed MoS₂ as shown below,

Supplementary Fig. 1. | Asymmetry of the density of states in deformed MoS₂. **a**, Doping charge density near the conduction band edge with the energy range of 0.3 eV. **b**, Doping charge density near the valence band edge with the energy range of 0.3 eV. The electron-doping concentration in (a) and (b) is $13.8 \times 10^{13} \text{ e/cm}^2$. The isosurface of charge density in (a) and (b) is 10^{-4} e/\AA^3 . **c**, Integrated DOS near the band edges for the intrinsic case of the deformed MoS₂. The integration of DOS was performed from the Fermi level to each band side. The valence band edge shows a larger capacity than that of conduction band edge. The yellow shade shows the difference between conduction and valence bands near the band edges. The energy is an absolute value. The zero energy is the Fermi level.

In line 20 on page 7, we have added a text below

“The asymmetry of the flattening between conduction and valence band edges were induced by different orbital configurations and capacities of LDOS in the curvature regions (Supplementary Fig.1).”

5) In Fig. 3, the authors show the spatial variation of the tunneling barrier height (TBH) and its correlation with surface morphology. However, many things need to be clarified in their plots and analysis. First, the unit of TBH must be eV, as shown in Fig. 3h.

Author Reply: The unit of the TBH is eV. We have corrected the unit in the legend and caption of Fig. 3a and e.

Second, how is the TBH surprisingly constant at the high curvature region?

Author Reply: Basically, the accumulated charge via doping screens potentials of each atom, which, eventually, results in the flattening of the local work function. The approximately constant value of TBH in curvature region in Fig. 3b and e was also predicted by theoretical results in Fig. 3h.

Initially, preference of filling sites is distinct between atoms, but accumulated charges start to screen the potential of each atom which makes an equilibrium of local work functions. The screening effects is dependent on capacity of occupation at a position. The electron-doping charge firstly prefers the orbital states of Mo atoms, which lowers the local work function on Mo atom positions further in the preference by curvature effects. (The higher electron density exhibits smaller local work function.) Therefore, Mo atom positions shows relatively smaller local work function than that of S atoms as shown in +1e case of Fig. 3h. When the higher electron doping is applied as shown in +3e case of Fig. 3h, the small peak features of relative variation in the curvature region shift from S to Mo atom positions due to the full major occupation of Mo sites. After the full filling of Mo atoms via electron doping, the doping charge prefers the sulfur sites, which results in a smaller local work function than that of Mo sites. Eventually, the amplitude of the small peak features becomes smaller by the occupation of S atoms when the higher doping is applied as shown in +3e case of Fig. 3h, indicating the leveling of preference between S and Mo. During the doping process, the charge screening by doping charge flattened the local work function for both the small peak features and their background in the curvature regions, which agrees with the experimental results. The difference in between electron and hole doping cases is mainly due to the difference of orbital configurations and capacity of LDOS in the curvature regions as discussed for issue 4 (Supplementary Fig. 1).

Third, the curvature plot in Fig. 3d must show the negative value on the hill like their extended data Fig. 3.

Author Reply: The ‘absolute’ values of (mean) curvatures are the main causes of band gap fluctuations. The curvature of hill and valley regions in Fig. 3d are all positive since Fig. 3d shows average values of the absolute curvatures. Each average value was obtained within the range of 12 nm as shown in Fig. 3a and b.

We have corrected the axis label in Fig. 3d to ‘absolute curvature’. In addition, we have added the following sentence in the caption of Fig. 3. “The average values for plots were obtained in the axis of 12 nm in (a) and (b).”

Taking the reviewer’s comments into account, we have presented the overall plot for each line profile of curvatures in “Supplementary Fig. 2”. In this plot, each line profile of curvature has opposite sign for the hill (–) and valley (+) regions.

Supplementary Fig. 2. | Total curvature plot from Fig. 3a. Each line profile of curvature in Fig. 3a were plotted. The average value of the absolute curvatures was given in Fig. 3d.

Last, they should have described the details of how to measure TBH.

Author Reply: In line 14 on page 17, we have added the more details for TBH in Methods.

“An apparent barrier height (TBH) is defined as $TBH = \frac{\hbar^2}{8m_e} \left(\frac{d\ln I}{ds} \right)^2$ where s (in Å) is the distance between sample and tip, m_e is the electron mass, \hbar is the reduced Planck constant, and the unit of TBH is given in eV. The ΔTBH is defined by difference between TBH and minimum of TBH (*i.e.*, $\Delta TBH = TBH - \text{minimum of TBH}$). The TBH was taken at the distance of ~ 6 Å between sample and tip (*i.e.*, $s = \sim 6$ Å).”

6) The authors analyzed the energy-dependent LDOS maps with autocorrelation functions and normalized histograms. Other people also used a similar way to prove the quantum criticality of metal-insulator transitions with the power-law behavior (Fig. 4e and f) and the log-normal distribution (Fig. 4g and h). To make their main idea more convincing, I strongly recommend showing the results of the “neutral” and “hole-doped” MoS₂. If they saw the differences or similarities, it would be great to explain them.

Author Reply: Following the reviewer’s comment into account, we have added the autocorrelation and multifractality in “hole-doped” case and description for “neutral” case in “Supplementary Fig. 5” as below,

“Supplementary Fig. 5. | Autocorrelation and multifractality in hole-doping case. **a**, LDOS maps for localized states near VBM and CBM (sample bias: -0.25 to 2.5 V) and extended state (sample bias: -1 V) in the hole-doping case of Fig. 1m. **b**, Radial autocorrelation results of Fig. 1m. The Fermi level is set to zero. **c**, Singularity spectra for LDOS maps for the hole-doping case. The extended state (-1 V) is close to the metallic limit (the narrow region near $\alpha = 2$ satisfying with $f(\alpha = 2) = 2$) due to the flattening of the valence band edge involving with narrower band tail width than that of conduction band edge. The localized states (-0.25 to 2.5 V)

exhibit stronger multifractality that the peak positions of singularity spectra are off-centered from $\alpha = 2$. **d**, Normalized LDOS distribution ($\text{LDOS}/\overline{\text{LDOS}}$) for the extended state (-1 V) shows a peak position near 1 of $\text{LDOS}/\overline{\text{LDOS}}$ indicating a uniform distribution. For the localized states, strong localization shows strongly skewed log-normal distributions of LDOS. If the hole-doping concentration becomes higher, the valence band edge will be more uniform exhibiting a narrower band tail width near the valence band edge and sharper LDOS distribution near its average value ($\text{LDOS}/\overline{\text{LDOS}} = 1$) following a normal distribution. In addition, the neutral (intrinsic) state of structural-disordered MoS_2 shows an even band tail width for the valence and conduction band edges by curvature formations, fast decaying behaviors in autocorrelation (short localization/correlation length), and strong multifractality due to the disorder-induced inhomogeneity of LDOS (Fig. 1d-f) without the flattening of band edges via doping.”

In line 24 on page 12, we have added the sentence;

“The experimental results for the hole-doping case (Fig. 1m) in autocorrelation, multifractality, and normalized distribution of STS agree well with the results of the electron-doping case (Supplementary Fig. 5).”

7) The authors argued that the randomly curved atomic structure (shown in STM images) leads to band tails. And their tight-binding calculations display the DOS spectra in Fig. 4i. It is natural to assume that their samples on SiO_2 have an inhomogeneous surface structure. The authors can categorize the dI/dV spectra based on the shape of band tails and make a correlation with the STM image.

Author Reply: For the details, we have added that STM/STS results for the formation of band tails that dependent on surface roughness or curvature in “Supplementary Fig. 4” as below;

Supplementary Fig. 4. | Band tail formation in structural disorder. Rough MoS_2 surface with higher curvatures shows larger band tail width in comparison with the band tail in the flat region. $\langle dI/dV \rangle$ indicates the spatial average of dI/dV results corresponding to the density of states. The flat region (denoted as ‘flat’) shows a larger band gap with negligible band tail widths, which is close to the intrinsic property of monolayer MoS_2 . The band tails of the regions labeled ‘1’ and ‘2’ in the insets exhibit significantly different widths. The average values of mean curvatures in the ‘1’ and ‘2’ regions are 0.1593 and 0.1867 nm^{-1} , respectively. Each curvature value corresponds to the bending strain of 2.632 and 3.111%. The higher value of curvature induces a more protruded band tail and larger band tail width. The arrows indicate protrusions of band tails.

In line 4 on page 13 of the manuscript, we have inserted the sentence that;

“... and experimental result are in Supplementary Fig. 4”.

8) The authors only discussed the structural disorder effects on the LDOS of a 2D semiconductor MoS_2 , but the SiO_2 substrate includes many impurities leading to the inhomogeneous electronic structure in the 2D material. How do they exclude this possibility?

Author Reply: We could identify defects and charged impurities with STM/STS results, therefore we ruled out their effects in the analysis of the curvature-induced effects.

Charged impurities act like an extrinsic potential inducing band bending, where both band edges change in the same direction. However, in curvature-induced band edge fluctuations, each band edge changes in the opposite direction, which differs from the effects of charged impurities. In our observations, charged impurities show unique features in STS spectra. The feature sizes of charged impurities are small (~a few nanometers) and sharp. The STS spectra affected by negatively charged impurities show the rapid upward band bending that exhibits a large change

in the local Fermi level position compared to the conduction band edge (below figures g-i). Such change in the local density of states is rapid and sharp, which differs from the smooth change of the flattening of band edges via doping. By the distinguishable features of charged impurities, we could rule out the effects of charge impurities in the analysis of the curvature effects and charge localization.

In addition, the defects' features in STM/STS results are in the size of ≤ 2 nm, which is far smaller than the typical scales of the curvature effects. Most defects involve additional states or in-gap states, which differentiates the defect effects from the curvature effects.

Characterization of defects and charged impurities. **a-c**, Atomic resolution STM images with defects in MoS₂ on SiO₂. The defects show the bright and dark spot features in the size of 1-2 nm scales. (sample bias: -3 V (**a**) and -2 V (**b,c**)) **d-e**, A LDOS line profile for a defect (**d**) shows an in-gap state and band gap reduction along with the surface morphology (**e**). The defect exhibited the small feature on the hill in the height profile of (**e**) and the curvature of the hill (the radius of curvature of 9.2 nm, indicated by a red-dashed line) induced the band gap reduction. In the figure (**d**), the rigid downshift of band bending near the narrow range of defect corresponds to a positive charge at the defect, which is different from the curvature-induced band edge changes that show the opposite directions in changes of each band edge (i.e., lowering of CBM and up-lifting of VBM). The results show overlapping of the effects from curvature (long range) and defect (short range). **f**, A single STS spectrum of the defect in (**e**). The arrow indicates the in-gap state of the defect. **g-h**, Observation of charged impurities. STM (**g**) and corresponding STS mapping (**h**) results for electron-doped MoS₂ on SiO₂ with charged impurities. The STM image (**g**) shows additional adsorbed particles (indicated by arrows). (sample bias: -3.5 V) The corresponding STS mapping results (**h**) at the sample bias of 1 V show dark regions exhibiting rigid shifts of the STS results on the axis of bias. **i**, STS results at the numbered spots in (**h**). The '0' spot did not involve any effects of charged impurities. The '1'-'4' spots exhibited the shifting of dI/dV spectra by the extrinsic potentials of charged impurities. The '1' and '2' spots included the adsorbed particles on MoS₂, but the '3' and '4' spots were affected by interfacial charged impurities between MoS₂ and SiO₂ surface. **j,k**, DFT calculations for the charged impurities. The negative (positive) charge type of impurities was modeled by a proximal effect of O (K) atom. By the potential of the negatively (positively) charged impurity, band bending shows an upward (downward) rigid shift near the charged impurity as shown in (**j**) ((**k**)). Such dramatic band bending by extrinsic potentials is different from the flattening of band edges via doping. The change of band edges by the extrinsic potentials occurred in the same direction, which is different from the curvature-induced band gap fluctuation that involves opposite directions in changes of band edges (i.e., lowering of CBM and up-lifting of VBM). By the distinguishable features of defects and charged impurities, we ruled out the defects and charged impurities from the analysis of curvature-induced band gap fluctuations and charge localization.

In line 15 on page 5 of the manuscript, we have inserted the sentence that;

“In the STM/STS results, the curvature-induced effects were clearly distinguished from the defects and charged impurities.”.

9) In Fig. 5, the authors showed that the curved atomic structure leads to a peculiar magnetic structure. Do their experimental results exist to support this theoretical expectation?

Author Reply: The intrinsic magnetism of monolayer MoS₂ on SiO₂ was already reported in the previous report (ref. 38), however, the origin of the intrinsic magnetization was not yet understood. Our DFT results of structural models align well with a previously reported experimental value of $\sim 4.3 \times 10^{-7} \mu_B/\text{\AA}^3$ and prove that the origin of the intrinsic magnetism is the structural disorder. Moreover, the result also agrees well with the average of Gaussian curvature($\kappa(\mathbf{r})$)-induced pseudo-magnetic fields from the STM result ($\sim 4.0 \times 10^{-7} \mu_B/\text{\AA}^3$), which supports the curvature-induced intrinsic magnetism further. (The average of curvature-pseudo-magnetic field relation: $\langle M \rangle = (\chi_{\text{MoS}_2}/\mu_0) \langle \mathcal{B} \rangle = (\chi_{\text{MoS}_2}/\mu_0) \langle \frac{\hbar}{2|e|} \kappa(\mathbf{r}) \rangle$, see Supplementary Discussion.). In addition, the previous study reported that the curvature-induced pseudo-magnetic field reproduced the experimental observation of the peculiar negative magnetoconductance in monolayer MoS₂.

On the other hand, if one considers the defect states as a source of magnetism, the observed defects in STM/STS results did not correspond to the previous results of magnetization. Most defects' configurations were predicted to be non-magnetic. Mo substitution with S and Mo vacancy are possible sources for the magnetization but they were not observed in the STM/STS results.

Taking the reviewer's comment into account, we have added further discussion in the manuscript,

Line 28 on page 15:

“Furthermore, the calculated magnetoconductance in the curvature-induced pseudo-magnetic field agrees well with the previous experimental results of the negative magnetoconductance in monolayer MoS₂ (ref. 45).”

Additionally, we have made changes on minor points to improve the manuscript further that;

In line 19 on Page 5, we have added “a-d” that “... results are presented in Extended Data Fig. 4a-d.” in the manuscript.

In line 6 on page 10, we have added “($\Delta\Phi$)” that “... show that the range of local work function fluctuations relative to the mean value ($\Delta\Phi$) increases ...” in the manuscript.

In line 28 on page 11, we have added “(See the Methods.)”.

In line 29 on page 11, we have modified “power law” to “power-law”.

In line 3 on page 12, we have added “for the criticality”.

In line 24 on page 13, we have added the reference 39 “... (ref.^{27,39}).”.

In line 4 on page 16, we have deleted “without long range order” for clarity.

In line 29 on page 17, we have added the sentence “(Supplementary Fig. 6a)” in Methods.

In line 1 and 6 on page 19 in Methods, we have modified “angle” to “radial”.

In line 36 on page 22 in References, we have added “*Nat. Commun.* **10**. 1584 (2019)”.

In Fig. 1, we have added the legends; “intrinsic (neutral)” for (d-g), “electron-doping” for (j), and “hole-doping” for (m).

In the caption of Fig. 4, we have modified “power law” to “power-law”.

We have corrected missing number indices for Extended Data Fig. 7c and added in caption of Extended Data Fig. 7, “the” in line 10 on page 38.

In line 3 on page 39 in the caption of Extended Data Fig 8, we have modified “Solid” to “A solid”.

In line 3 on page 39 in the caption of Extended Data Fig 9, we have added “a”.

We have modified that, Extended Data Fig. 10 changes to Extended Data Fig. 9, and for Extended Data Fig. 9 to Extended Data Fig. 10, reversely.

All the changes in the manuscript have been remarked by red color.

REVIEWER COMMENTS

Reviewer #1 (Remarks to the Author):

the authors have addressed the questions raised. I recommend the publication of the manuscript as it is

Reviewer #2 (Remarks to the Author):

In the revised manuscript, authors have well addressed my concerns. They have shown more explanation on the underlying mechanisms and more discussion and comparison with other possible defects. I am satisfied with the current version and recommend for publication.

Reviewer #3 (Remarks to the Author):

In their response to the first review and the revised manuscript, the authors have effectively addressed several concerns raised during the review process through their plausible arguments. Their supplementary materials strengthen their ideas by providing detailed information on how the physical quantities were derived from the data and eliminating the possibility of atomic impurities affecting the spatial variation of the z-height contrast. However, there are still some concerns that remain.

First, the experimental data still lacks evidence to support the authors' claims regarding the magnetism of strained 2D MoS₂. While previous reports may have allowed the authors to estimate the pseudo-magnetic field's strength based on the autocorrelation decay rate (ν) of the autocorrelation and the gaussian curvature, their data does not directly demonstrate the spatial variation of local magnetic moments. I also don't see the point of discussing magnetism in the main text. How is this related to the main message the authors try to convey? It might be more suitable to shift it and related statements to supplementary information.

Second, to enhance the reliability of their analysis, it is recommended that the authors include additional spectra corresponding to curvature values between 0.1593 and 0.1867 1/nm in their supplementary figure 4.

For this paper to be considered for publication in Nature Communications, I recommend a restructuring to emphasize the central idea and the presentation of all relevant information in a comprehensive manner.

Below is the 'Point-by point response' to the reviewer' comments,

REVIEWER COMMENTS

Reviewer #1 (Remarks to the Author):

the authors have addressed the questions raised. I recommend the publication of the manuscript as it is

Reviewer #2 (Remarks to the Author):

In the revised manuscript, authors have well addressed my concerns. They have shown more explanation on the underlying mechanisms and more discussion and comparison with other possible defects. I am satisfied with the current version and recommend for publication.

Reviewer #3 (Remarks to the Author):

In their response to the first review and the revised manuscript, the authors have effectively addressed several concerns raised during the review process through their plausible arguments. Their supplementary materials strengthen their ideas by providing detailed information on how the physical quantities were derived from the data and eliminating the possibility of atomic impurities affecting the spatial variation of the z-height contrast. However, there are still some concerns that remain.

First, the experimental data still lacks evidence to support the authors' claims regarding the magnetism of strained 2D MoS₂. While previous reports may have allowed the authors to estimate the pseudo-magnetic field's strength based on the autocorrelation decay rate (ν) of the autocorrelation and the gaussian curvature, their data does not directly demonstrate the spatial variation of local magnetic moments.

Author Reply: The data presented in the manuscript were produced in three completely independent methods (see below). As it is written in the manuscript, all those results are surprisingly well aligned. Although there is no direct experimental value of pseudo-magnetic field's strength, we believe the agreement of all existing results, including the results of three independent theoretical approaches based on our experimental values and results of previously published papers are strong enough to support our claim.

1. The **experimentally** measured surface morphology was used to evaluate the pseudo-magnetic field B_{Gauss} , by using the theoretical formulation of curvature effects.
2. The theoretical ν value was evaluated by the theoretical formulation of ν in a system with random spin-orbit interactions in a weak magnetic field B .

As a function of magnetic field B , the theoretical $\nu(B)$ value at the given $B_{Gaussian}$ coincided with the fitting value of ν from the autocorrelation results of the STS results.

3. In addition, we have characterized the localization/correlation lengths from the radial-averaged autocorrelation profile of the STS results, independently, and the exponent ν was extracted by fitting of $\sim|E-E_C|^{-\nu}$, which evaluated ν of 2.73 in good agreement with the contour fitting of the radial-averaged autocorrelation. (The fitting result is below.)

Supplementary Fig. 7 | Characterization of localization/correlation length near the critical energy.

The radial-averaged autocorrelation profiles (Fig. 4e and See Methods.) were characterized by fitting localization/correlation length of equation, $\sim \exp(-|\mathbf{R}|/\xi)$ where $|\mathbf{R}|$ is the radial distance and ξ is the localization/correlation length. To obtain the critical exponent ν from the localization/correlation lengths from the above, the equation of $\sim |E - E_C|^{-\nu}$ was fitted. The obtained critical exponent value is 2.73 which coincides to the fitting results of the contour of the radial-averaged autocorrelation profile (Fig. 4e) and the theoretical values from the curvature-induced pseudo-magnetic field (Extended Data Fig. 9) as described in the main manuscript. All the results from independent approaches agreed well with each other, suggesting that the curvature mechanism of electronic/spin structure played an important role.

I also don't see the point of discussing magnetism in the main text. How is this related to the main message the authors try to convey? It might be more suitable to shift it and related statements to supplementary information.

Author Reply:

The magnetization is a critical issue to understand the symmetry class of metal-insulator transitions in structurally disordered MoS₂. The symmetry class is strongly related to the criticality of a system, and known to define whether metal-insulator transition is allowed or not.

Especially, spin-orbit interactions are related to the special symmetry class, called symplectic class. For the symplectic class, random variations in the spin-orbit interaction break the spin-rotation symmetry and maintain the time-reversal symmetry, enabling the metal-insulator transition. For MoS₂, spin-orbit interaction is an essential factor to understand its electronic structures. Since the spin-orbit interaction is associated with the net in-plane dipole that is strongly related to the local structures, the curvature-induced structural disorder in MoS₂ forms a random distribution of the spin-orbit interaction.

On the other hand, when the time-reversal symmetry is broken (as in the unitary class), metal-insulator transition is expected to be prevented. Interestingly, the intrinsic magnetization of MoS₂ (non-zero value) violates the time-reversal symmetry, which means the metal-insulator transition is not expected. The prediction of unitary classification is, therefore, contradicted to the observations of metal-insulator transitions in MoS₂. It implies that another approach in consideration of symmetries is needed for the metal-insulator transition in MoS₂. Even in the previous results, such symmetry classification could not satisfy all the previous experimental results from various systems, which suggested that it was not sufficient to predict the criticality of a system. Researchers developed other approaching methods with interacting pictures and supersymmetry on the nonlinear sigma model in a viewpoint of quantum field theory, however it cannot solve the problem completely. Therefore, this is an issue that needs to be further thought out and developed. In that point of view, we believe the understanding of the magnetization in relation with structural disorder is critical and should be presented in the main manuscript.

To make this point clearer,

In line 10 on page 15 in the manuscript, we have modified the subtitle as

“Symmetry classes and structural-disorder-driven magnetism”

In line 12 on page 15 in the manuscript, we have added

“The local variations in SOC should be investigated in a viewpoint of symmetry classes.”

In line 12 on page 16 in the manuscript, we have added

“On the other hand, fitting value of ν obtained from the localization/correlation lengths characterized in the radial-averaged autocorrelation profile was 2.73 (Supplementary Fig. 7). All the results of ν from independent approaches coincided with each other.”

In line 12 on page 17 in the manuscript, we have added

“Direct measurements of local magnetization in atomic scale are needed to understand the criticality of quantum fluctuations further, using spin-polarized STM or magnetic force microscope.”

and more results for localization/correlation length near the critical energy E_C in “Supplementary Fig. 7”

Second, to enhance the reliability of their analysis, it is recommended that the authors include additional spectra corresponding to curvature values between 0.1593 and 0.1867 1/nm in their supplementary figure 4.

Author Reply: We have added additional results of STS mapping from different surface roughnesses of MoS₂ on SiO₂. All the experimental results were in good agreement with the theoretical results in Fig. 4i. The larger curvature distribution in MoS₂ exhibited the larger band tail width near the band edges. Each averaged STS value contained over 10⁴ data points, which is large enough for average values to track the trend of band tail widths as a function of curvature.

Supplementary Fig. 4 | Band tail formation in structural disorder. Rough MoS₂ surface with higher bending curvatures shows larger band tail width in comparison with the band tail in the flat region. $\langle dI/dV \rangle$ indicates the spatial average of dI/dV results corresponding to the density of states. The flat region (denoted as ‘flat’) shows a larger band gap with negligible band tail widths, which is close to the intrinsic property of monolayer MoS₂. The band tails of the regions labeled ‘1’ to ‘4’ in the side insets exhibit significantly different widths. The average values of mean curvatures in the ‘1’, ‘2’, ‘3’, and ‘4’ regions are 0.1593, 0.1763, 0.1867, and 0.1995 nm⁻¹, respectively. Each curvature value of ‘1’ to ‘4’ corresponds to the bending strain of 2.632, 2.912, 3.111, and 3.3589%, respectively.

The higher value of curvature induces a more protruded band tail and larger band tail width, which is in good agreement with the theoretical calculations in Fig. 4i. The arrows indicate protrusions of band tails.

For this paper to be considered for publication in Nature Communications, I recommend a restructuring to emphasize the central idea and the presentation of all relevant information in a comprehensive manner.

Author Reply: Considering the reviewers comment, we have added several sentences (see page 1 and 2) to deliver our main idea more clearly.

Minor corrections for clarity:

We have modified the sentence with “... by fitting of the contour in Fig. 4e ...” in line 9 on page 16 in the manuscript.

We have added “On the other hand, the ...” in line 15 on page 15 in the manuscript.

We added the missing affiliation “⁴Department of Nano Science and Technology, Sungkyunkwan University (SKKU), Suwon 16419, Republic of Korea”.